# Review on Preventive Measures to Reduce Post-Weaning Diarrhoea in Piglets

**DOI:** 10.3390/ani12192585

**Published:** 2022-09-27

**Authors:** Nuria Canibe, Ole Højberg, Hanne Kongsted, Darya Vodolazska, Charlotte Lauridsen, Tina Skau Nielsen, Anna A. Schönherz

**Affiliations:** Department of Animal and Veterinary Sciences, Aarhus University, Blichers Allé 20, 8830 Tjele, Denmark

**Keywords:** antimicrobial resistance, feeding strategies, medical zinc, piglets, post-weaning diarrhoea

## Abstract

**Simple Summary:**

Post-weaning diarrhoea (PWD) is a major challenge in pig production, which results in the high use of ZnO for its prevention and the utilisation of antibiotics for its treatment. There is a global agenda to decrease the use of antibiotics in order to reduce the risk of antimicrobial resistance (AMR); moreover, from June 2022, administering medical levels of ZnO has been banned in many countries due to its negative impact on the environment and AMR. Many feeding strategies and feed additives have been found to be tools to reduce PWD risk, which affect the gut microbiota and the host through different modes of action. Many feeding strategies and additives show positive but variable effects on PWD. A combination of such interventions tailored according to the specific conditions is most probably the best strategy.

**Abstract:**

In many countries, medical levels of zinc (typically as zinc oxide) are added to piglet diets in the first two weeks post-weaning to prevent the development of post-weaning diarrhoea (**PWD**). However, high levels of zinc constitute an environmental polluting agent, and may contribute to the development and/or maintenance of antimicrobial resistance (**AMR**) among bacteria. Consequently, the EU banned administering medical levels of zinc in pig diets as of June 2022. However, this may result in an increased use of antibiotic therapeutics to combat PWD and thereby an increased risk of further AMR development. The search for alternative measures against PWD with a minimum use of antibiotics and in the absence of medical levels of zinc has therefore been intensified over recent years, and feed-related measures, including feed ingredients, feed additives, and feeding strategies, are being intensively investigated. Furthermore, management strategies have been developed and are undoubtedly relevant; however, these will not be addressed in this review. Here, feed measures (and vaccines) are addressed, these being probiotics, prebiotics, synbiotics, postbiotics, proteobiotics, plants and plant extracts (in particular essential oils and tannins), macroalgae (particularly macroalgae-derived polysaccharides), dietary fibre, antimicrobial peptides, specific amino acids, dietary fatty acids, milk replacers, milk components, creep feed, vaccines, bacteriophages, and single-domain antibodies (nanobodies). The list covers measures with a rather long history and others that require significant development before their eventual use can be extended. To assess the potential of feed-related measures in combating PWD, the literature reviewed here has focused on studies reporting parameters of PWD (i.e., faeces score and/or faeces dry matter content during the first two weeks post-weaning). Although the impact on PWD (or related parameters) of the investigated measures may often be inconsistent, many studies do report positive effects. However, several studies have shown that control pigs do not suffer from diarrhoea, making it difficult to evaluate the biological and practical relevance of these improvements. From the reviewed literature, it is not possible to rank the efficacy of the various measures, and the efficacy most probably depends on a range of factors related to animal genetics and health status, additive doses used, composition of the feed, etc. We conclude that a combination of various measures is probably most recommendable in most situations. However, in this respect, it should be considered that combining strategies may lead to additive (e.g., synbiotics), synergistic (e.g., plant materials), or antagonistic (e.g., algae compounds) effects, requiring detailed knowledge on the modes of action in order to design effective strategies.

## 1. Introduction

Post-weaning diarrhoea (**PWD**) in pig production is a widespread and worldwide disease with high morbidity, consequently leading to productivity loss and mortality. PWD is considered to be one of the main disorders contributing to the use of antibiotics and administering medical levels of zinc oxide (**ZnO**) in pig production [1]. Antibiotics and medical levels of ZnO have a detrimental impact on human health and the environment by contributing to the development of antimicrobial resistance (**AMR**) among bacteria and to a high zinc—which is a heavy metal—concentration in the soil [2,3]. Studies have shown that medical doses of dietary zinc oxide increase the tetracycline and sulphonamide resistance genes of Gram-negative bacteria in weaned pigs [4,5]. The World Health Organization considers AMR to be a global health and development threat, and the misuse and overuse of antimicrobials are the main drivers in the development of drug-resistant pathogens [6].

Therefore, reduction in the use of antibiotics in pig production has been a recent scientific focus. The use of antibiotics as growth promoters (**AGPs**) was banned in the EU in 2006, and several countries worldwide have followed; the search for strategies to reduce the level of antibiotics is ongoing. In the years after the ban on the use of AGP, the usage of ZnO by adding medical levels (2500 ppm Zn as ZnO) to the feed during the first two weeks post-weaning to prevent PWD increased in some countries, including Denmark [7], and the levels are still high [8].

As mentioned above, ZnO has a negative impact on public health, due to the link between high dietary zinc levels and occurrence of AMR [3,4], as well as negative effects on the environment [2]. Jensen et al. [2] concluded that ‘the current use of zinc and copper in pig production may lead to leaching of metals, especially zinc, from fields fertilised with pig slurry in concentrations that may pose a risk to aquatic species’. Therefore, on 26 June 2017, the European Commission adopted a decision to withdraw all marketing authorisations for veterinary medical products containing zinc oxide administered orally to food-producing species, effective from June 2022 [9]. This has resulted in an intensive search for nutritional and management strategies that can contribute to preventing PWD in the absence of medical levels of ZnO, in order to avoid an increase in the use of antibiotics after June 2022. In this search, a large number of strategies have been tested and proposed. In this review, we focus on the key strategies which have been studied, and some which are more novel, but that seem to have good perspectives.

In this review, we focus on diarrhoea occurring during approximately the first two weeks post-weaning, because this is the period during which medical ZnO is provided today. Thus, the emphasis will be studies that include diarrhoea/faecal score as a response parameter, and that investigate the immediate period after weaning, mostly in the first two weeks.

Although PWD is a multifactorial disease [10,11,12], enterotoxigenic *E. coli* (**ETEC**) expressing the fimbriae F4 and F18 are considered as main etiological agents [11,13]; therefore, challenge studies with these two bacteria are models often used to investigate the impact of strategies to prevent PWD; these type of studies, as well as non-challenge studies, are included here. On the other hand, no association between the detection of pathogens (ETEC F4 and F18, *Lawsonia intracellularis* and *Brachyspira pilosicoli*) and diarrhoea status of individual pigs, or between the detection of pathogens in a pen and diarrhoea floor pools has been reported [14]. This suggests that factors other than pathogens can lead to diarrhoea, and an immediate reflection from these observations is that antibiotics are most probably not always the appropriate treatment to diarrhoeic piglets. These aspects are highly relevant to discuss, and further research is needed, but this is beyond the aim of this review.

The field of alternative strategies to the use of antibiotics (and medical level of ZnO) is very broad and a very large amount of literature is available; therefore, some priorities were made in this review.

Some specific strategies (among many) not included in the current review are presented subsequently.

Management strategies: although undoubtedly highly relevant, these were not considered in this review. Here, only feeding strategies and vaccines were included.

Reduced dietary crude protein level: this strategy is already effectively applied in practice. Reducing the level of dietary crude protein reduces the risk of suffering from PWD [15,16,17,18]. A possible risk of this practice is that growth performance can be negatively affected, but the addition of synthetic amino acids to comply with recommendations can be added to avoid reduced productivity [16]. Furthermore, some data indicate that when considering a longer period of time and not only the immediate weeks after weaning, no differences in growth between pigs fed a diet with standard dietary protein levels or a diet with low dietary protein levels without extra amino acid supplementation are observed [16,19,20]. Nevertheless, in order to avoid detrimental effects on performance, it is important to optimise the dietary protein level, i.e., not too low, and the application period in the specific situation, i.e., a balance between minimizing the period of feeding with low protein diets and reducing PWD.

Organic acids: these additives are widely added to weaner diets and considered to have a beneficial impact against PWD, as reviewed by Lopez-Galvez et al. [21] and Heo et al. [17]. They will only be briefly included here.

The aim of the current review was to present and evaluate the main feeding strategies and vaccines that can be considered as alternatives to the administration of medical ZnO and antibiotics during the immediate post-weaning period to avoid PWD in piglets, with the exclusions and focuses mentioned above.

## 2. Search Strategy

An initial search was based on articles published in peer-reviewed journals in the database Web of Science. Articles and reviews written in English were selected. The literature was retrieved through an electronic search for articles published between 1999 and 2021. Relevant scientific papers were identified using the keyword combinations (piglet or piglets or pig or pigs or weaner or weaners or porcine or swine) AND (weaning or postweaning or post-weaning or post weaning) AND (diarr*). In total, 1547 articles were found. This database was then used by all the authors of this review, who then added the specific words of the strategies to be included. Relevant articles found in the reference lists of these articles were also included.

## 3. Prevention of Post-Weaning Diarrhoea—Mode of Action

A plethora of studies have been conducted over the years, with a wide range of feeding strategies aiming at avoiding/reducing the levels of antibiotics and medical zinc used in pig production in relation to PWD. The strategies include changes in feed ingredients or their levels (e.g., protein level and source, soluble/insoluble fibre, and fat composition), feed additives (e.g., probiotics, prebiotics, enzymes, antimicrobial peptides, organic acids, plant extracts, and amino acids), and feed form (e.g., liquid, dry, coarse, fine, pelleted, and meal). Other strategies which have been studied to a lower extent but with promising perspectives include vaccines, bacteriophage treatment (phage therapy), and single-domain antibodies. The mode of action by which these strategies exert their effect can be different; therefore, their relevance can also vary depending on the specific challenges identified in different production sites.

The main modes of action behind the effect of most widely studied strategies to prevent PWD can be classified according to the following (Table 1).

(1)Establishment of a robust gut microbiota

The gut of a newborn pig is sterile, but very quickly becomes colonised by a relatively low number of microbial species, and diversity increases within the first weeks of life. Colonisation of the digestive tract is a progressive process in which some microorganisms, although able to colonise the digestive tract, are supplanted by others better adapted to life in the changing gastrointestinal ecosystem of the developing animal [22,23,24,25].

The dynamics of this early colonisation is characterised by common patterns, i.e., facultative anaerobic and aerotolerant species colonise initially, and as the gut environment turns more anaerobic, strict anaerobic species proliferate and a more diverse microbiota is established [22,23,24,25]. The dynamics of this early microbiota succession is an important factor that influences the risk of suffering from microbial disturbances leading to PWD, i.e., it impacts the resilience of the host during the critical period around weaning.

In human research, it has been shown that babies born by caesarean section compared with vaginal birth have a higher risk of suffering from allergic disease [26,27], a possible contributing factor being a different composition of the first gut colonisers and a lower diversity [27]. One study with piglets showed that animals which developed PWD had a different faecal microbiota composition, evenness, and diversity pre-weaning compared with those without PWD [28], which indicates that microbiota composition before weaning can have influence the susceptibility to PWD. Identifying the composition of this ‘preventive’ microbiota is a challenge that has not yet been solved.

Several feeding strategies investigated for piglets fall within this mode of action, e.g., the addition of probiotics, prebiotics, and synbiotics to modify the microbiota composition. The treatment should be applied very early in the piglet’s life (as soon as possible after birth); therefore, the addition of the additive to the feed of the dam represents a convenient and feasible method [29,30,31,32,33]. The rationale behind this practice is that by providing the dam with ingredients/additives that improve the composition of her gut microbiota, the piglets will be exposed to this beneficial microbiota from birth in the vaginal passage and in the pen environment. In principle, every strategy that has a beneficial impact on the dam’s gut microbiota would beneficially affect the early colonisation of the offspring. Therefore, feed ingredients such as fibre source and content could also be included here. There are also some studies reported in the literature in which probiotics have been provided directly to the piglets shortly after birth [34,35]. This strategy is hypothesised to increase the chance of modulating the gut microbiota to improve gut health/robustness to disease. In these studies, the piglets were individually provided with specific probiotics several times after birth attempting to promote their colonisation or to beneficially influence microbiota establishment, even though the probiotics may not establish themselves.

(2)Promoting maturation of the gut before weaning

One of the most significant abrupt changes piglets are subjected to at weaning is the change in feed composition and consistency. The transition from the sow’s milk to a dry and mainly plant-based feed results most often in a period of low feed intake or anorexia. In practice, suckling pigs are offered creep feed, most often dry, and less often as liquid feed, to let them become familiar to this form of feed and as a nutritional supplement to the sows milk for those piglets not being able to suckle enough [36]. Furthermore, and importantly, it is presumed that creep feed intake contributes to maturation of the gut function by stimulating the development of the gastrointestinal tract (**GI-tract**) microbiota and the activity of digestive enzymes (e.g., pancreatic enzymes and brush border enzymes) necessary to hydrolyse substrates of plant origin [36,37,38]. This would mean that when piglets are weaned, their digestive system is more developed and more capable of digesting and absorbing dietary nutrients than piglets not ingesting any creep feed, and that their gut microbiota can better withstand the disturbances occurring at weaning. Studies aiming at testing this hypothesis have shown equivocal results [38,39,40]. One important challenge of this strategy is that piglets consume little creep feed in the first three weeks of life, with a more significant intake only in the fourth week of life [41]. Moreover, there is a percentage of animals that do not consume a significant amount of creep feed [38,42]. Therefore, focus has been directed towards strategies that increase the intake of plant-based feed pre-weaning.

Providing feed with a consistency closer to that of milk, i.e., liquid feed instead of dry feed, is considered to help the animals during the weaning transition, and thereby hypothesised to partially prevent the reduction in feed intake post-weaning [38,43,44]. Another strategy is to modulate the gut development and function of the offspring via the dam, which has been demonstrated by providing antibiotics to the sows as a proof of concept [45], and by feeding with different fibre sources [46].

(3)Inhibition of/reduction in the growth of pathogens (antibacterial effect)

As described above, some of the main etiological agents of PWD are various strains of ETEC, mainly those expressing F4 and F18 fimbria. Therefore, an obvious strategy to reduce PWD is assumed to be reducing the level of these bacteria.

Strategies such as the addition of organic acids, plant components with antibacterial effects, antimicrobial peptides including bacteriocins, single-domain antibodies (nanobodies), bacteriophages, probiotics, fibre, and strategies that reduce the stomach/small intestine lumen pH, such as fermented liquid feed and coarsely ground feed, could be candidates to inhibit or reduce the growth of ETEC. Some of these strategies are already used in pig production and others are being studied as potential candidates to prevent/treat PWD. Aiming to optimally obtain an antibacterial effect implies pursuing a targeted antibacterial effect, i.e., the intention is to impede the growth of ETEC without affecting the growth of beneficial bacteria, such as lactic acid bacteria (**LAB**). This principle is clear in the case of bacteriophages, which have high specificity. In the case of antimicrobial peptides, the specificity varies depending on the peptides in question [47]. It is also suggested that fibre sources w resemble host receptors might interrupt the adherence of bacteria to the intestinal mucosa, thereby reducing *E. coli* colonisation and proliferation in the small intestine by blocking the *E. coli* adhesion to the gut epithelial cells [48,49,50]. The other strategies mentioned may not specifically affect the growth of ETEC/coliforms, but act by reducing the digesta pH, for example. Coliforms are less tolerant to low pH than LAB [51]; therefore, the result is a greater reduction in the number of coliforms than that of LAB. In accordance, data from studies using these additives and feeding strategies have shown that the growth of coliforms is reduced to a higher extent than that of LAB, for example [51,52,53].

Another aspect considered important is that the strategies should reduce the level of ETEC/coliforms already in the stomach. The theory behind this is that the stomach can act as a barrier. Thus, if the pathogens are killed in the stomach, fewer will pass to the small intestine, where they may otherwise colonise and cause disease [53]. At the same time, fewer pathogens will be shed in the faeces, thereby reducing the load of horizontal pathogen transmission among pigs.

(4)Promotion of the growth of beneficial bacteria (and indirect reduction in pathogens)

In addition to directly killing specific pathogens or hampering their growth, another strategy to reduce the proliferation of pathogens/*E. coli* is to promote the growth of beneficial bacteria, and thereby indirectly, via several possible mechanisms, reduce or impede the growth of pathogens. Feeding probiotics, prebiotics, and synbiotics are some strategies in this group. Thus, either the beneficial microorganism itself is provided via the feed (probiotics), or substrates are supplied to stimulate the growth of the targeted beneficial microorganisms in the gut (prebiotics).

Once the beneficial microorganisms proliferate in the gut, the techniques by which they reduce/impede the growth of pathogens are considered to involve one or several of the following mechanisms [54,55,56,57]:–Reduction in the pH by the production of fermentation products, e.g., lactic acid;–Competitive exclusion, i.e., inhibition of the attachment of pathogens by the physical obstruction of attachment sites;–Production of antibacterial substances such as bacteriocins, hydrogen peroxide, antibiotics, and organic acids;–Nutrient competition;–Modulation of the immune response;–Inactivation of enterotoxins;–Maintenance of homeostasis via quorum sensing.(5)Modulation of the immune response/provision of immune protection

One factor considered crucial in affecting the risk of the individual host to suffer from PWD is its immune competence and how it is affected by the weaning process. The presence of commensal bacteria has a direct influence on the immune maturation, and the microbial exposure influences the expression of a large number of immune-related genes [58]. Hence, host-independent factors in connection with weaning, such as nutrition and the rearing environment, which influence the gut microbiota, may also influence immune activation. Nutritional stress in terms of anorexia post-weaning coupled with its consequences on gut morphology have been reported to be major contributors to local intestinal inflammation during the immediate post-weaning period [59]. Additionally, ETEC secretes toxins and produces virulence factors that exploit host cell functions to facilitate bacterial colonisation. Many of these bacterial proteins subvert the inflammatory response of the host cell. These mechanisms include pro- and anti-inflammatory responses that favour bacterial survival and growth [60].

Several additives, ingredients, or strategies can be expected to exert their beneficial action by modulating the immune response, e.g., milk replacers, probiotics, immunoglobulins, vaccines, antimicrobial peptides, plants/plant extracts, algae, amino acids, etc. In fact, most strategies, directly or indirectly, can have an impact on the immune response of the host.

As illustrated in the above paragraphs, one additive/strategy can exert its action through more than one mode of action, which will depend on the gut ecosystem in the individual host at the time of application.

## 4. Strategies to Prevent/Reduce Post-Weaning Diarrhoea

Some considerations must be made when reviewing and concluding on the impact of alternatives to antibiotics and medical ZnO on PWD and the associated parameters.

There can be a bias regarding published findings, resulting in an over-representation of studies with positive results.

Studies investigating the impact of strategies on diarrhoea often register the faecal score as a measure of diarrhoea. In some cases, although the faecal score has been reported to be improved by the alternative tested, i.e., firmer consistency of faeces, none of the pigs (including the control pigs) suffered from diarrhoea. In these cases, although the finding is positive in principle, the impact of the alternative on PWD is not really proven.

Many studies include a small number of animals, which makes the results less robust.

The statistical methods are not always described in detail.

### 4.1. Probiotics

Probiotics are live microorganisms which, when administered in adequate amounts, confer a health benefit on the host [61,62]. Probiotics can contain spore-forming (*Bacillus* spp. and *Clostridium* spp.) and non-spore-forming (all others) bacteria, as well as non-bacterial microorganisms such as yeast, formulated as single- or multi-strain complexes. Microorganisms utilised as probiotic additives must be non-pathogenic, non-toxic, and free of transmissible antibiotic resistance genes [63]. To promote health, probiotics must furthermore be able to survive and grow in the GI-tract. Hence, probiotics must be able to tolerate the harsh manufacturing, transportation, storage, and application processes, and resist gastric acid, bile salts, and pancreatic enzymes, in order to colonise the intestinal tract [60,64]. To date, many probiotics are commercially available for supplementation into pig diets. The most common probiotics used in pig production belong to the genera *Bacillus*, *Bifidobacterium*, *Clostridium*, *Enterococcus*, *Lactobacillus*, and *Pediococcus*, whereas the most common non-bacterial probiotics belong to the species *Saccharomyces (S.) cerevisiae* [21].

#### 4.1.1. *Bacillus*

*Bacillus*-based probiotics have obvious advantages because they are spore-forming, which makes them thermostable for feed storage and processing (i.e., pelleting and extrusion) and facilitates survival at low pH in the stomach [65]. Several studies have demonstrated that dietary supplementation with probiotic Bacillus strains could improve growth performance, reduce the incidence of diarrhoea, and improve gut morphology in weaned pigs [66,67,68,69]. Moreover, members of the Bacillus genus have also been reported to produce bacteriocins, which are proteins with antimicrobial properties that can inhibit the activity of pathogenic bacteria [70,71].

Dietary supplementation with *B. subtilis* (10^9^ CFU/kg feed) was shown to improve growth performance, mildly alleviate diarrhoea severity, enhance gut health (greater crypt depth, elongated villi height, and increased goblet cell numbers), and reduce the systemic inflammation of weaned pigs orally infected with ETEC F18, whereas supplementation with *B. pumilus* (1 × 10^9^ CFU/kg feed) mainly alleviated systemic inflammation, but showed only a limited impact on growth performance and PWD [72] (Table 2). In accordance, Hu et al. [68] reported that probiotic treatment with *B. subtilis* KN-42 (2 × 10^9^, 4 × 10^9^, or 20 × 10^9^ CFU/kg feed) had a positive impact on the incidence of diarrhoea, particularly within the first 14 days post-weaning, where the effects of probiotics on PWD were comparable with in-feed antibiotics (such as neomycin sulphate). *B. subtilis* KN-42 supplementation further improved the bacterial diversity of the intestinal environment, increased the relative number of Lactobacillus, and reduced the relative amount of *E. coli* shed in the faeces. In a study by Dumitru et al. [67], supplementation with *B. licheniformis* at much higher doses, i.e., 1.6 × 10^9^ CFU spores/g feed or 4.8 × 10^9^ CFU spores/g feed, significantly reduced the incidence of PWD by 40% and 55.5%, respectively. Moreover, *B. licheniformis* reduced *E. coli* numbers both in GI-tract contents and faeces, promoted the growth of beneficial microbiota (*Lactobacillus* spp. and *Bacillus* spp.) in the ileum and cecum, and positively influenced the pH of gut contents. In line, Lin and Yu [69] showed that a *B. licheniformis*-fermented feed additive, alone (1 g/kg *B. licheniformis*-fermented feed additive) or in combination with a reduced dose of the antibiotic bacitracin, reduced the incidence of PWD from day 1 to day 14 post-weaning.

In contrast, probiotic treatment with *B. subtilis* DSM 25841 (1.28 × 10^9^ CFU/kg feed, or 2.56 × 10^9^ CFU/kg feed) only improved growth performance and gut barrier function, but did not show positive effects on diarrhoea scores or frequency in ETEC-challenged weaners [73], nor PWD in unchallenged weaners following *B. subtilis* supplementation (1 × 10^5^ or 1 × 10^6^ CFU/kg feed) [74], whereas the combined use of *B. licheniformis* and *B. subtilis* neither improved the PWD nor growth performance of weaners challenged with *E. coli* K88ac [75]. Similarly, supplementation with *B. amyloliquefaciens* (7.5 × 10^5^ CFU/g feed) could not alleviate PWD nor the negative impact on growth performance and gut health following F18 ETEC challenge in weaned piglets [48].

**Table 2 animals-12-02585-t002:** Effects ^a^ of probiotics on various parameters in piglets.

Treatment	Microorganism/s	Dose	Pathogenic Challenge	Endpoints ^b^								Refs.
				GP	ND	FT/D	FS	GA	IS	FA	GM	
** *Bacillus* **												
Single-strain	*Bacillus* (*B.*) *longum* subsp. *infantis* CECT 7210	1 × 10^9^ CFU/day	2 × 10^9^ and 6 × 10^9^ CFU of *Salmonella (S).* Typhimurium on days 8 and 10	NS	nm	NS	+	+	+	+	+	[76]
Single-strain	*B. longum* subsp. *infantis* CECT 7210	1 × 10^9^ CFU/day	5 × 10^9^ and 5 × 10^10^ CFU of *E. coli* K88 on days 5 and 6	NS	nm	NS	+	+	+	+	+	[77]
Single-strain	*B. licheniformis* ATCC 21424	L: 1.6 × 10^9^ CFU spores/g feed H: 4.8 × 10^9^ CFU spores/g feed	nm	NS	nm	+	nm	nm	nm	nm	+ (L)	[67]
Single-strain	T1: *B. subtilis* DSM 32540, T2: *B. pumilus* DSM 3253,	500 mg/kg feed (1 × 10^9^ CFU/kg)	1 × 10^10^ CFU of ETEC F18, 3 consecutive days	+ (T1)	nm	+ (T1)	nm	+	+	nm	+/−	[72]
Single-strain	*B. subtilis* KN-42	L: 2 × 10^9^ CFU/kg feed, M: 4 × 10^9^ CFU/kg feed, H: 20 × 10^9^ CFU/kg feed	nm	NS (1–14 dpw), + (M,H: 1–28 dpw)	nm	+	nm	nm	nm	nm	+	[68]
Single-strain	*B. amyloliquefaciens* SC06	1 × 10^9^ CFU/ kg feed	nm	+	nm	+	nm	nm	nm	nm	nm	[78]
Single-strain	*B. licheniformis*-fermented feed additive	1.0 g/kg (5 × 10^8^ CFU/kg feed	nm	NS	nm	+	nm	+	nm	nm	+	[69]
Single-strain	*B. subtilis* GCB-13-001	L: 1 × 10^5^ CFU/kg feed H: 1 × 10^6^ CFU/kg feed	nm	+	NS	NS	+ (PRO2)	nm	nm	nm	nm	[74]
Multi-strain	T1: 3 strains *B. amyloliquefaciens,* T2: *T1 and B. subtilis*	T1: 7.5 × 10^5^ CFU/g feed, T2: 1.5 × 10^5^ CFU/g feed	6 mL ETEC F18+ (~1.9 × 10^9^ CFU/mL) at 7dpw	NS	nm	NS	T1: NS, T2: − (2 dpi), + (7 dpi)	NS	NS	Nm	nm	[48]
Multi-strain	*B. licheniformis* (DSM 5749) and *B. subtilis* (DSM 5750)	L: 3.9 × 10^8^ CFU/day, M: 7.8 × 10^8^ CFU/day, H: 3.9 × 10^9^ CFU/day	*E. coli* K88ac (O149:K91, F4+), 1 × 10^10^ CFU/day	NS	nm	NS	nm	nm	nm	Nm	+	[75]
Multi-strain	Spray-dried spore-forming *B. licheniformis* (DSM 5749) and *B. subtilis* (DSM 5750)	L: 10 mL 3.9 × 10^7^ CFU/mL, H: 10 mL 7.8 × 10^7^ CFU/mL	10 mL *E. coli* K88ac (O149:K91, F4+), 1.0 × 10^9^ CFU/mL	+	nm	NS	nm	nm	+/−	nm	nm	[79]
Single-strain	*B. subtilis* (DSM 25841)	L: 1.28 × 10^9^ CFU/kg feed H: 2.56 × 10^9^ CFU/kg feed	3 mL F18 *E. coli* (1 × 10^10^ CFU)	+ (H)	nm	NS	nm	+ (H)	NS	nm	nm	[73]
**Lactic acid bacteria/Bifidobacteria**												
Single-strain, multi-species (T3)	T1: *Lactobacillus (L.) casei,* T2: *Enterococcus (E.) faecalis,* T3: *L. casei and E. faecalis* (3:1)	1–4 mL, 1 × 10^9^ CFU/mL	nm	+	nm	+	nm	+ (only reported for T3)	+ (only reported for T3)	nm	+ (only reported for T3)	[80]
Single-strain	T1: *L. plantarum,* T2: *L. reuteri,* T3: *L. plantarum and L. reuteri*	2 × 10^8^ CFU/g	nm	T1: NS, T2 and 3: -	nm	+	nm	nm	nm	nm	nm	[81]
Single-strain	*L. plantarum* JC1 (B2028)	2 × 10^10^ CFU/day	6 mL (2 × 10^9^ CFU/mL) of *E. coli* K88 (O149:K91:H10 /LT-I/STb)	+	nm	NS	nm	+	+	(+)	+	[82]
Single-strain	*E. faecalis* LAB31	L: 0.5 × 10^9^ CFU/kg feed, M: 1 × 10^9^ CFU/kg feed, H: 2.5 × 10^9^ kg feed	nm	+	nm	NS (1-14 dpw), + (H: 1–28 dpw)	nm	nm	+ (H)	nm	+ (H)	[83]
Single-strain	*L. rhamnosus* (heat-killed)	1 × 10^9^ FU/g feed L: 0.1% M: 0.2% H: 0.4%	Nm	+	+ (H)	+	nm	nm	+	nm	nm	[84]
Single-strain	*L. plantarum* CJLP243	L: 1 × 10^8^ CFU/kg feed M: 1 × 10^9^ CFU/kg feed H: 1 × 10^10^ CFU/kg feed	5 × 10^9^ CFU of *E. coli* K88ac	+	nm	+	+	nm	+	nm	nm	[85]
Single-strain	*L. rhamnosus*	L: 10 mL/day of 10^9^ CFU/mL, H: 10 mL/day of 10^11^ CFU/mL	*E. coli* K88ac (O149:K91, F4+), 1.6 × 10^9^ CFU/mL, 10 mL/day	nm	nm	+	+	nm	+	nm	+	[86]
Single-strain	*Bifidobacterium (Bif.) lactis* HN019	10 mL of 1 × 10^8^ CFU/mL	nm	nm	nm	+ (1 - 3 dpw)	nm	+	nm	nm	nm	[87]
Single-strain	T1: *L. plantarum* Zj316, T2: *L. plantarum* Zj316 + antibiotic	T1L: 1 × 10^9^ CFU/day, T1M: 5 × 10^9^ CFU/day, T1H: 1 × 10^10^ CFU/day, T2: 1 × 10^10^ CFU/day	nm	+ (T1L)	nm	+	nm	+ (only presented for T1L)	nm	NS	NS	[88]
Single-strain	*L. plantarum* PFM105	2 × 10^7^ CFU/g feed	Nm	+	nm	+	nm	+	nm	+	+	[89]
Single-strain	*E. faecium* DSM 10663 NCIMB 10415	Newborn: 2.8x10^9^ CFU, suckler: 2.52 × 10^9^ CFU/day, weaner with PWD: 2.9-5.8x10^8^ CFU/day	nm	+	nm	+	nm	nm	nm	nm	nm	[90]
Single-strain	*L. rhamnosus* GG	10 mL, 1 × 10^10^ CFU/mL	10 mL of *E. coli* K88ac (O149:K91, F4+), 1 × 10^9^ CFU/mL	nm	nm	+	+	nm	+	nm	+	[91]
Single-strain	*E. faecium* R1	6.5 × 10^6^ CFU/g feed	intraperitoneally injected lipopolysaccharides (*E coli* O55:B5, 100 μg/kg body weight)	NS (+ F/G ratio)	nm	+	nm	nm	nm	nm	nm	[92]
Multi-strain	*Bif. longum* subsp. *infantis* CECT 7210 *Bif. animalis* subsp. *lactis* BPL6	1 × 10^9^ CFU/day	5 × 10^8^ CFU *S.* Typhimurium	NS	nm	+	+	+	+	+	nm	[77]
Multi-strain	*L. reuteri* and *L. plantarum*	L: 0.1% 1 × 10^9^ CFU/kg feed H: 0.2% 1 × 10^9^ CFU/kg feed	nm	+ (L: 0–28 dpw)	+ (L)	+ (L)	+ (L)	nm	nm	nm	+	[93]
Multi-species	*L. murinus* DPC6002 and DPC6003, *L. salivarius* subsp. *Salivarius* DPC6005, *L. pentosus* DPC6004, and *Pediococcus pentosaceous* DPC6006; T1: as fermented milk, T2: as milk suspension	T1: 4 × 10^10^ CFU/day, T2: 4 × 10^9^ CFU/day	1 × 10^8^ CFU/day *S. enterica* Typhimurium PT12 for three consecutive days	NS	nm	+	+	nm	nm	nm	nm	[94]
Multi-species	*L. acidophilus,* *L. casei,* *Bif. thermophilum,* *E. faecium*	0.25 × 10^8^ CFU/g feed of each strain	*E. coli* S1191 and 2144 (F18^+^; 1 × 10^9^ CFU/mL of each strain)	+	nm	ns	nm	+ (jejunum)	+	nm	nm	[95]
** *E. coli* **												
Single-strain, multi-strain (T4)	T1: *E. coli candidates* 582 T2: *E. coli candidates* B771 T3: *E. coli candidates* B1172 T4: *E. coli candidates* 582, B771 and B1172 combined	5 mL/day, 1 × 10^9^ CFU/mL	ETEC 12919 and ETEC 173 (1 × 10^6^ CFU/mL)	nm	nm	+ (T4)	+ (T4)	nm	nm	nm	NS	[96]
Multi-strain	T1: *E. coli* UM-2 and *E. coli* UM-7; T2: T1 + raw potato starch	3.1 × 10^9^ CFU/mL	6 mL *E. coli* K88+ (3 strains, 2.3 × 10^9^ CFU/mL)	+	nm	+ (T2)	+	nm	nm	+ (T2)	+	[97]
**Yeast**												
Single-strain	*Saccharomyces (S.) cerevisiae* CNCM I-4407	1 × 10^10^ CFU/g, 5 × 10^10^ CFU/g (weaner)	*E. coli* (O149:K88, LT+), 1.5 × 10^11^ CFU/pig	+ (42 dpw)	nm	+	+	nm	+	nm	nm	[98]
Single-strain	*S. cerevisiae* CNCM I-4407	T1: 5 × 10^10^ CFU/kg (0–21 day) T2: 5 × 10^10^ CFU/kg (7–11 day) T3: 2 × 10^11^ CFU/kg (once)	*E. coli* F4ac, 1 × 10^8^ CFU	NS	nm	+ (12 h, 48 h)	+ (d4)	nm	NS	nm	m	[99]
Single-strain	T2: *Candida utilis* (*C.*)*,* T2: *C. utilis* + Yucca Schidigera extract (YSE)	T1: 1 × 10^9^ CFU T2: 1 × 10^9^ CFU CU + 120 mg YSE/kg feed	nm	+ (T1)	nm	+	nm	+	nm	nm	+	[100]
Single-strain	*S. boulardii* mafic-1701	1 × 10^8^ CFU/kg	nm	NS (+ F/G ratio)	nm	+	nm	nm	+	+	+	[101]
Single-strain	T1: Duan-Nai-An (*S. cerevisiae* S288c fermented egg white; combination of live yeast and active egg white) T2: *S. cerevisiae* S288c	10 mL/day (2.0 × 10^8^ CFU/mL)	nm	nm	nm	+	nm	+ (T1)	nm	nm	nm	[102]
**Multi-species**												
Multi-species	*T1: E. faecium* DSM7134 (1 × 10^8^ CFU/g), *B. subtilis* AS1.836 (2 x 10^10^ CFU/g), *S. cerevisiae* ATCC 28338 (1 × 10^10^ CFU/g) T2: *E. faecium* DSM7134 (1 × 10^8^ CFU/g), *B. subtilis* AS1.836 plus (2 x 10^10^ CFU/g), *L. paracasei* L9 CGMCC (1 × 10^10^ CFU/g)	1g probiotic/kg feed	nm	+	nm	+	nm	nm	nm	+	+ (T2)	[103]
Multi-species	probiotic: *B. mesentericus* TO-A (10^8^ CFU/g), *Clostridium butyricum* TO-A (10^8^ CFU/g), *E. faecalis* T- 110 (10^10^ CFU/g); T1: maternal + neonatal administration; T2: neonatal administration	maternal: 0.2% (*w*/*w*), neonatal: 0.02% (*w*/*w*)	nm	+	nm	+	nm	+ (T1)	NS	nm	+ (T2)	[104]
Multi-species	*B. licheniformis* and *S. cerevisiae*	500 mg/kg	10 mL *E. coli* K88+ (1 × 10^9^ CFU/mL)	+	+	+	nm	+	nm	nm	+	[105]
Multi-species	T1: *E. faecium* 6H2, *L. acidophilus* C3, *Pediococcus pentosaceus* D7, *L. fermentum* NC1 T2: T1 + *B. subtilis* H4, T3: T2 + *S. boulardii* Sb	2 mL/kg feed of each probiotic strain (1.3 × 10^9^–8 × 10^11^ CFU/mL)	nm	+	+	+	nm	nm	nm	+	+	[106]

^a^ Effects +: positive effect; -: negative effect; NS: non-significant effect; nm: not measured; ^b^ GP: growth performance; ND: digestibility; FT/D: faecal traits/reduction in diarrhoea; FS: faecal pathogen shedding (after exp. challenge); GA: gut architecture; IS: immune system; FA: fermentative activity; GM: gut microbiota.

#### 4.1.2. *Bifidobacterium*

Bifidobacteria are Gram-positive, non-sporing, non-motile, lactic-acid-producing bacteria that naturally inhabit the GI-tract of pigs. Although bifidobacteria have exhibited antagonistic activity against *E. coli*, the number of studies investigating *Bifidobacterium* probiotics as alternative strategies in preventing PWD in pig is limited. Shu et al. [87] demonstrated that supplementation with *B. lactis* HN019 (10^9^ CFU/piglet) via an orogastric tube has the potential to reduce the severity of weanling diarrhoea associated with rotavirus and *E. coli.* Reduced levels of infection in the probiotic-fed piglets, as indicated by reductions in the faecal shedding of rotavirus and *E. coli*, were associated with higher blood leukocyte phagocytic and T-lymphocyte proliferative responses and higher GI-tract-pathogen-specific antibody titres, suggesting enhanced immune-mediated protection as a possible mechanism. Dietary supplementation of a multi-strain probiotic, such as *B. longum* subsp. *Infantis* CECT 7210 or *B. animalis* subsp. *Lactis* BPL6 (1 × 10^9^ CFU), had a positive effect on reducing pathogen loads and ameliorated the outcome of a *Salmonella* Typhimurium challenge using a weaner piglet model [77]. Administration of the multi-strain probiotic improved faecal consistency, with significant decreases in faecal scores in both challenged and non-challenged pigs. Beneficial effects observed in probiotic-fed piglets included decreased diarrhoea scores, reduced faecal shedding of Salmonella, decreased rectal temperatures, improved fermentation profiles on day 8 post-infection (increased ileal acetic acid and a tendency to lower colonic ammonia concentrations), an improved villous:crypt ratio, and an increased voluntary feed intake. On the other hand, supplementation of the same *B. longum* subsp. *infantis* CECT 7210 strain alone (1 × 10^9^ CFU), following a similar experimental design by the same group, also diminished *Salmonella* shedding but failed to show significant improvements in clinical outcomes [76].

#### 4.1.3. *Enterococcus*

Zhang et al. [92] demonstrated that dietary supplementation with *E. faecium* R1 (6.5 × 10^6^ CFU/g feed) not only improved the intestinal function of piglets by decreasing the incidence of diarrhoea, but also alleviated intestinal and liver injury in piglets challenged with LPS. In line, dietary supplementation with *E. faecalis* LAB31 (0.5 × 10^9^ CFU/kg feed, 1 × 10^9^ CFU/kg feed, 2.5 × 10^9^ CFU/kg feed) improved growth performance, reduced diarrhoea, and increased the relative number of Lactobacillus in the faeces of weaned piglets [83]. Beneficial effects were dose-dependent and most pronounced in high-dose-administrated pigs. A study by Zeyner and Boldt [90] revealed that the daily oral supplementation of *E. faecium* DSM 10663 NCIMB 10415 (1.26 × 10^9^ CFU) to piglets from birth to weaning twice a day through drenching reduced the percentage of piglets that suffered from diarrhoea and improved their daily weight gain. On the other hand, it was shown in the same study that a glucose-based solution with an additional 2.9–5.8 × 10^8^ CFU of *E. faecium* did not have any therapeutic effects once diarrhoea was present. However, the improvements in diarrhoea scores and the percentage of viable piglets that developed diarrhoea after *E. faecium* supplementation lead to the conclusion that the probiotic stabilised the gut environment, which later translated into improved daily weight gain.

#### 4.1.4. *Lactobacillus*

Supplementation with *L. plantarum* ZJ316 in drinking water was shown to improve growth performance and gut health, and was able to alleviate PWD more effectively than dietary antibiotics [88]. Moreover, probiotic effects were dose-dependent, revealing that supplementation with 1 × 10^9^ CFU/day of *L. plantarum* ZJ316 resulted in more pronounced effects than higher doses (5 × 10^9^ CFU/day or 1 × 10^10^ CFU/day). The authors suggested that the observed probiotic effects might be related to the growth inhibition of opportunistic pathogens and the promotion of increased villus height along the GI-tract. Comparable results were observed by Wang et al. [89], reporting that supplementation with lyophilized *L. plantarum* PFM105 (2 × 10^7^ CFU/g feed) was more effective than in-feed antibiotics in alleviating PWD. Revealing positive effects on gut health and immune and inflammatory responses, dietary supplementation with *L. plantarum* JC1 (2 × 10^10^ CFU/day) in weaned piglets challenged with *E. coli* K88 only marginally reduced the incidence of diarrhoea (*p* = 0.09) and did not affect the performance, nor *E. coli* K88 counts, in the colon [82]. Dietary supplementation with *L. plantarum* (2 × 10^8^ CFU/g feed) and *L. reuteri* (2 × 10^8^ CFU/g feed) individually or in combination (1 × 10^8^ CFU/g *L. plantarum* plus 1 × 10^8^ CFU/g *L. reuteri*) significantly reduced the diarrhoea occurrence and faecal scores but did not influence pig performance, nor total faecal bacteria, faecal lactobacilli, and coliform counts [81]. In particular, the occurrence of PWD was lowest in *L. plantarum*-treated pigs, whereas multi-strain-treated pigs exhibited the lowest (firm faeces) faecal scores over time. Effects observed using the multi-strain probiotic were in general agreement with findings by Zhao and Kim [93] using the same probiotics.

Oral supplementation with 10 mL of 1 × 10^10^ CFU *L. rhamnosus* GG ATCC 53103 after experimental infection with *E. coli* K88 reduced the incidence of diarrhoea, lowered faecal coliform counts, and increased lactobacilli counts. It was shown that *L. rhamnosus* led to increased concentrations of secretory IgA in the jejunum and ileum, and high titres of TNF-α in addition to reduced diarrhoea scores [91]. In contrast, Trevisi et al. [107] reported that the dietary supplementation of weaner diets with 6 × 10^9^ CFU *L. rhamnosus* GG ATCC 53103 had neither preventative nor control properties on adverse effects in pigs challenged with enterotoxigenic *E. coli* O149: F4ac. Li et al. [86] reported that a high dose of *L. rhamnosus* (1 × 10^11^ CFU/mL, 10 mL/day) in piglets negated the preventive effect against ETEC F4 compared with the administration of a lower dose of *L. rhamnosus* (1 × 10^9^ CFU/mL, 10 mL/day). The authors suggested that high doses of certain probiotics may negate the preventative effects, at least in part by disturbing the established microbial ecosystem and by interfering with mucosal immune responses against potential enteric pathogens.

#### 4.1.5. *Escherichia coli*

*E. coli* is a commonly occurring enteric bacteria in pigs, exhibiting a lifestyle that ranges from that of an obligate pathogen to a commensal, indicating that not all *E. coli* strains are pathogenic or disadvantageous. In fact, *E. coli* Nissle 1917 has been used in Europe as a licensed probiotic drug in humans for chronic inflammatory, functional, and infectious bowel diseases for almost 90 years [108]. *E. coli* has been shown to produce two types of bacteriocins allowing for the direct antimicrobial inhibition of competing bacteria [109,110,111]. More than half of the *E. coli* strains isolated from human faecal microbiota [109,112], as well as pathogenic *E. coli* strains isolated from pigs [113], produce at least one type of bacteriocin. An additional benefit of selecting probiotic strains of the same genus and species as the pathogen to be excluded is that the organisms typically occupy the same niche in the gut, and hence, compete for the same ecological niche and resources. A disadvantage of using *E. coli* as a probiotic is that it is not generally considered safe.

The use of bacteriocinogenic *E. coli* to inhibit porcine ETEC strains was found to be an effective approach both in vitro [114] and in vivo [96,97,115]. In the study by Hrala et al. [96], three potentially probiotic *E. coli* strains of human origin (582, B771, and B1172) were shown to exert probiotic effects on porcine pathogenic ETEC under both in vitro and in vivo conditions. Following the administration of 1 × 10^9^ CFU per dose of each *E. coli* strain individually or in combination, piglet diarrhoea caused by an ETEC challenge and pathogenic *E. coli* shedding were reduced, suggesting synergy between the tested probiotic *E. coli* candidates. Moreover, only the group treated with a cocktail of probiotic *E. coli* had piglets that did not suffer from any type of diarrhoea. These findings were in agreement with a previous study by Schroeder et al. [116], reporting that pre-treatment with probiotic *E. coli* Nissle 191 was shown to decrease clinical signs of diarrhoea following ETEC challenge. However, the study had a very low number of replicates (only four animals in the probiotic-treated group). In the study by Krause et al. [97], only the combination of probiotic *E. coli* (UM-2 and UM-7) and raw potato starch had a beneficial effect on growth performance and resulted in a reduction in diarrhoea following pathogenic *E. coli* K88 challenge, whereas the effects observed for the probiotic treatment alone were less pronounced. However, the study lacked a negative control group because the control group (*E. coli*-K88-challenged without inclusion of raw potato starch or probiotic *E. coli*) included in-feed antibiotics. Although promising, studies investigating the effects of bacteriocinogenic *E. coli* on PWD in pigs are limited. Moreover, the effects are only moderate when applied as single-strain probiotics, but improve when applied as multi-strain cocktails or combined with prebiotics. Specific modes of action of *E. coli* probiotics still need to be elucidated.

#### 4.1.6. Yeast

Zhang et al. [101] reported that weaners supplemented with dietary *S. boulardiimafic-1701* (1 × 10^8^ CFU/kg feed) showed higher feed efficiency in the last 14 days and a lower diarrhoea rate over the entire experimental period of 28 days. The authors concluded that the improved feed conversion ratio and reduced diarrhoea rates following *S. boulardiimafic-1701* supplementation may have been associated with enhanced antioxidant activity, anti-inflammatory responses, and improved intestinal microbial ecology. In line with these findings, Trckova et al. [98] reported that ETEC-challenged pigs receiving *S. cerevisiae* var. *boulardii* showed a significant improvement in faecal scores, the duration and severity of diarrhoea, and the shedding of ETEC in faeces accompanied by increased growth performance compared with control pigs. The dietary supplementation of Duan-Nai-An (2 × 10^8^ CFU/mL, 10 mL/day), a yeast culture of the tamed *S. cerevisiae* on egg white, significantly improved weight gain and feed intake, and reduced diarrhoea and mortality rates of early-weaned piglets [117]. The gut bacterial community was significantly shaped by Duan-Nai-An, whereas the fungal gut community was not affected. In addition, Zhaxi et al. [102] demonstrated that Duan-Nai-An (2 × 10^8^ CFU/mL, 10 mL/day) as a dietary supplement helped to maintain and improve the morphology and structure of mucosal epithelial cells as well as the integrity of the intestinal mucosa in the small intestine of weaning piglets. In line, dietary supplementation with *Candida utilis* (1 × 10^9^ CFU/mL, 1 mL), alone or in combination with yucca schidigera extract, improved growth performance, reduced diarrhoea rates, improved animal antioxidant capacity by increasing T-SOD, T-AOC, and CAT activity, enhanced gut morphology and absorption capacity (increased ileal villus height, villus height/crypt depth), improved intestinal integrity by increasing the concentrations of jejunal occludin and β-definsin-2, and increased microbial diversity in the caecum of weaned piglets [100]. The effects on PWD, mortality, and improved antioxidant capacity were more pronounced when *C. utilis* and yucca schidigera extract were administered combined [100].

#### 4.1.7. Multi-Species Probiotics

Multi-species (strains belonging to different genera) probiotics have been suggested to have greater efficacy than single strains, because complementary or even synergistic effects can be achieved [118,119,120]. However, it has also been stressed that the greater variety of probiotic genera present may reduce its effectiveness through mutual inhibition by the different species, antimicrobial compounds, or competition for either nutrients or binding sites [118]. In pigs, multi-species probiotics have been demonstrated to alleviate PWD and improve general gut health when used as dietary supplements [80,103,104,106]. Several combinations have shown potential to mitigate the negative outcomes of pathogen challenges [77,79,94,105]. Unfortunately, only a few studies have compared the effects of the multi-species probiotics with those of the individual strains alone.

Lui et al. [80] demonstrated that supplementation with a probiotic complex containing *L. casei* and *E. faecalis* improved growth performance and reduced diarrhoea rates and mortality by improving gut health, immune status, and maintaining microbial balance during weaning transition. Health-promoting effects were more pronounced when fed as a complex compared with strains fed individually. Comparable results were reported when feeding weaned piglets with multi-species probiotics containing either *E. faecium* DSM 7134 (1 × 10^8^ CFU/kg feed), *B. subtilis* AS1.836 (2 × 10^10^ CFU/kg feed), and *L. paracasei* L9 (1 × 10^10^ CFU/kg feed), or *E. faecium* DSM 7134 (1 × 10^8^ CFU/kg feed), *B. subtilis* AS1.836 (2 × 1010 CFU/kg feed), and *S. cerevisiae* TCC 28338 (1 × 10^10^ CFU/kg feed) [107]. However, the separate effects of each strain were not investigated, limiting conclusions on the superiority of combining probiotics. In a study by Pan et al. [109], dietary supplementation with a multi-species probiotic containing 0.75 × 10^10^ CFU *B. licheniformis* and 0.15 × 10^10^ CFU *S. cerevisiae*/kg feed showed the potential to replace in-feed antibiotics in weaned pigs challenged with ETEC K88 (1 × 10^9^ CFU/mL). Both the antibiotic and probiotic treatment improved growth performance and decreased the incidence of diarrhoea. Differences between antibiotic and probiotic effects on PWD were not detected. Moreover, positive effects on gut morphology, growth of beneficial gut microbiota (Lactobacillus), and reduction in caecal *E. coli* counts were more pronounced in pigs fed the probiotic [105]. On the other hand, dietary supplementation with a multi-species probiotic containing *L. acidophilus*, L. casei, B. thermophilum, and *E. faecium* (0.25 × 10^8^ CFU/g feed for each strain) did not reduce the severity of PWD following ETEC challenge, but still enhanced growth performance, reduced intestinal inflammation and oxidative stress, and improved gut morphology [95]. In addition, oral treatment with a five-strain lactic acid bacteria probiotic complex (*L. murinus* DPC6002, *L. murinus* DPC6003, *L. salivarius* subsp. *salivarius*, *L. pentosus*, and *Pediococcus pentosaceous)* ameliorated diarrhoea in weaned piglets following challenge with *Salmonella enterica serovar* Typhimurium (1 × 10^8^ CFU) [94]. The probiotic complex was supplied fermented (4 × 10^10^ CFU/day) or as suspension (4 × 10^9^ CFU/day). The application of either of the probiotic treatments resulted in reduced numbers of faecal Salmonella and lowered the risk of diarrhoea. The addition of probiotic strains in fermented liquid feed, facilitating probiotic growth during the fermentation process, could be a technique for increasing the amounts supplied. This way, probiotics added to the liquid feed at doses recommended for dry feed (typically 10^9^ CFU/kg) could reach concentrations up to 10^9^ CFU/g feed—1000 times higher (Canibe et al., unpublished). However, the impact of this strategy on the piglets, including PWD, needs to be investigated.

Beneficial effects of multi-species probiotics are associated with various modes of action, including the competitive exclusion of pathogenic bacteria, modulation of gut microbiota, immunomodulation, and anti-oxidation. However, the effects of multi-species probiotics in practice are not consistent, and could be influenced by strain composition, dosage, formula, feeding environment, and the nutritional level of feed, as well as the age and health status of animals. Moreover, further research is required to obtain a better understanding of symbiotic or synergistic combinations to maximise the positive benefits of probiotic combinations.

#### 4.1.8. Limitations

A large amount of research has been conducted in recent years investigating the effect of a vast array of probiotics on pig health and growth performance, promoting probiotics as promising antibiotic replacers during the weaning transition. However, observed effects on PWD are often marginal, and in several studies, diarrhoea induced/observed in control treatments was only mild, complicating robust conclusions. It is important to realise that the effects of probiotics are treatment-specific, depending on the particular strain, dose, experimental setup, and duration [86,121]; as well as host-specific, depending on host-related physiological parameters (e.g., health status and genetics) and environmental conditions (e.g., sanitary status and diet) [122,123,124]. Conditions under which probiotics have been tested are far from standardised, and the use of different strains, diverse doses, and the effect of uncontrolled variables such as the age of pigs at the time of challenge, variations in treatment dose, buffer type and method of probiotic delivery, as well as the duration of the adaptation period or duration of the treatment period, may have introduced unwanted variability, limiting the power to retrieve robust and reproducible results [21,63]. In many of the studies reviewed, sample sizes were limited. Moreover, the recording of diarrhoea was typically based on faecal consistency scores, although the utilised scaling systems differed across studies. In addition, interventions comparing antibiotic- and probiotic-treated groups in the same trial are still sparse. Altogether, the huge variation in experimental conditions, detection methods, and markers measured to monitor probiotic effects on piglet health makes it difficult to compare results adequately and draw generalised conclusions on the use of probiotics as antibiotic replacements. As emphasised by Barba-Vidal et al. [63], it is important to recognise that probiotics should be considered health promoters and gut microbiota stabilisers rather than antimicrobial therapeutics. Instead, combining them with other feed and/or management strategies with a more holistic approach may be necessary. The mentioned limitations hold, in fact, to most of the alternatives described in this review.

Finally, manufacturing and application processes may impair the efficacy, viability, and desirable characteristics of in-feed probiotics [125], and on-farm storage conditions with variable temperatures, moisture, and long shelf-life conditions may further impair probiotic survival. Alternative strategies such as microencapsulation, microsphering, or dietary supplementation in fermented liquid feeds may hold great potential to enable non-spore bacteria to bypass feed manufacturing and storage constraints as well as protect them from gastric acid, bile salts, and pancreatic enzymes before reaching the intestine; thus, this should be further investigated to improve the efficacy of probiotics.

### 4.2. Prebiotics

Several definitions have been proposed for prebiotics in the past, but the recent one proposed by The International Scientific Association for Probiotics and Prebiotics (ISAPP) is: ‘a substrate that is selectively utilized by host microorganisms conferring a health benefit’ [126]. This definition expands the concept of prebiotics to possibly include non-carbohydrate substances, applications to body sites other than the gastrointestinal tract, and diverse categories other than food. Beneficial health effects must be documented for a substance to be considered a prebiotic. Although the impact on bifidobacteria and lactobacillus has been the focus so far, it is recognised today that prebiotic effects probably extend beyond bifidobacteria and lactobacilli; however, to meet the selectivity criterion of a prebiotic, the range of microorganisms affected must be limited [126]. A prebiotic, in addition to having a selective effect on microorganisms, must also evoke a net health benefit. A number of fermentable carbohydrates have been reported to convey a prebiotic effect, but the dietary prebiotics most extensively documented to have health benefits in humans are the non-digestible oligosaccharides fructans, galactans, and lactulose [126]. Substrates that affect the composition of the microbiota through mechanisms not involving selective utilisation by host microorganisms are not prebiotics. There still seems to be unclarity in what components are considered prebiotics.

Gibson et al. [126] mention prebiotics such as oligosaccharides of fructose, mannose, and chitin as having shown protection in piglets against high environmental stressors (such as antibiotics) and pathogen loads, including faecal *E. coli* shedding and reduced infection-associated responses to Salmonella enterica serovar Typhimurium infection.

The dietary supplementation of a chito-oligosaccharide product at 100, 200, or 400 mg/kg to piglets decreased the incidence of diarrhoea, and increased the faecal shedding of lactobacilli during three weeks after weaning. Furthermore, the 100 and 200 mg/kg doses increased small intestine villi height, apparent faecal digestibility, and growth performance [127] (Table 3).

Hossain et al. [128], feeding weaners with 0.5% and 1% lactulose, detected small but significant differences in lactobacilli and *E. coli* counts approximately 3 weeks after weaning, the former being higher and the latter lower when lactulose was added to the feed. No effect of lactulose addition on faecal score or faecal DM% was observed. The values did not indicate diarrhoea in the control group, which made it difficult to find an effect of the prebiotic. In an ETEC K88 challenge study with weaners, the impact of adding 1% lactulose was investigated [82]. The challenge promoted a mild-moderate diarrhoea, and no effect of lactulose was detected. Ten days post-challenge, but not six, the lactulose group had more lactobacilli and a higher molar ratio of butyric acid in the colon digesta compared with the control group. This was accompanied by a lower plasma concentration of the acute phase protein Pig-MAP (but not TNF-α) and increased ileal villi height.

Jensen et al. [129] detected a higher number of *Bifidobacterium* in the ileum of piglets fed with 30% chicory (containing high levels of inulin), but as in [128], no diarrhoea developed in any group. Similarly, Halas et al. [136], fed 4% or 8% inulin to weaners. PWD was only observed sporadically; therefore, no impact of inulin could be measured. No impact on ileal morphology was detected. Chen et al. [131] added 1% inulin to the diet fed the first two weeks post-weaning and found no effect on the incidence of diarrhoea. The study had a very low number of replicates, i.e., six per treatment, however.

Adding a prebiotic product containing β-glucans, glucomannans, and mannan-oligosaccharides (**MOS**) derived from the cell wall of the yeast *S. cerevisiae* to the diet at the levels of 1, 2, or 3 g/kg did not affect the incidence of diarrhoea in weaners when followed for 35 days post-weaning [132]. Other parameters were measured on day 35 post-weaning, considered too late in the context of this review (where the focus was the first ~two weeks). Faecal scores were not affected either by adding 0.1%, 0.2%, or 0.3% galactomannan oligosaccharides from sesbania gum, containing 20% galactose and 15% mannose [137]. On the other hand, adding 0.6% of an isomalto-oligosaccharide product containing isomaltose, panose, isomaltotriose, and other branched oligosaccharides composed of more than four glucose units to the diet very slightly but significantly improved faecal scores measured during the four weeks after weaning [133]. Piglets fed this product also had lower serum levels of malondialdehyde and higher of antioxidant enzymes on day 14, which was interpreted as an amelioration of the antioxidant status by the prebiotic. Similarly, adding a fructo-oligosaccharide (**FOS**) (0.1%), but not MOS (0.1%) or both (0.05% FOS + 0.05% MOS), slightly improved faecal scores the first week post-weaning [134]. The FOS and the mixture resulted in higher total tract digestibility of DM and nitrogen 14 days post-weaning [134]. In the study by Castillo et al. [135], 0.2% supplementation of an MOS derived from the outer cell wall of a selected strain of yeast (BioMos) improved faecal consistency only on day 7 when measured daily during the first three weeks post-weaning. The group fed the prebiotic had lower numbers of Enterobacteria in the jejunum, but no effect on lactobacilli counts, serum concentration of immunoglobulins, ileal IgA, number of intraepithelial lymphocytes, or goblet cells or villus height in the jejunum measured 14 days post-weaning. Using the same product and dietary inclusion level (0.2% BioMos), Valpotic et al. [138] reported a reduction in the number of diarrhoeic piglets when followed for 35 days post-weaning. According to Stuyven et al. [139], piglets (n = 5 or 6) fed with three β-glucans differing in origin (*S. cerevisiae* or *Sclerotium rolfsii*) for two weeks after weaning were less susceptible to an F4+ ETEC infection in comparison with a control group, as seen by a reduction in the faecal excretion of ETEC F4, reduced F4-specific serum antibody response, and lower diarrhoea severity. These differences were only significant at a few time points, however.

In summary, some studies described here did not detect diarrhoea in the control groups, and could therefore not test the impact of the prebiotics on this parameter. Other studies reporting significant differences in faecal score reflected some small changes in the faecal consistency without reaching diarrhoea in any of the groups. In several cases, the studies were designed so that the animals were slaughtered 4–5 weeks after weaning, which may be considered too late when dealing with PWD related mainly to ETEC and when searching for alternatives to administering medical levels of ZnO, i.e., the first two weeks post-weaning. These considerations limit the number of studies that qualify for the evaluation of the efficacy of adding prebiotics to weaner feed; therefore, although some positive results have been reported, the available data do not allow to consider prebiotics as an alternative to medical ZnO or antibiotics in order to prevent PWD. Although some of the products included in this section, i.e., MOS from *S. cerevisiae*, should be classified as postbiotics (see postbiotic section below), they have been included here because they are so far widely named prebiotics.

### 4.3. Synbiotics

The definition of synbiotic has been updated by the International Scientific Association for Probiotics and Prebiotics to ‘a mixture comprising live microorganisms and substrate(s) selectively utilised by host microorganisms that confers a health benefit on the host’ [140]. Two subsets of synbiotics have been defined: complementary and synergistic. A ‘synergistic synbiotic’ is a synbiotic in which the substrate is designed to be selectively utilised by the co-administered microorganism(s) [140]. A ‘complementary synbiotic’ is a synbiotic composed of a probiotic combined with a prebiotic, which is designed to target autochthonous microorganisms [140].

Three different synbiotics composed of *S. cerevisiae*, combinations of *Lactobacillus* spp. strains, and inulin were fed to sows from day 10 before farrowing and during the whole lactation, and to their piglets from weaning [141]. Positive correlations between the synbiotic groups and *Lactobacillus* sp. and *Bifidobacterium* sp., and lactic acid and some short chain fatty acids (**SCFAs**), were observed. On the other hand, negative correlations between synbiotics and *Clostridium* sp., *Enterococcus* sp., *Enterobacteriaceae* family, *E. coli*, and yeasts were reported. Feeding piglets the synbiotic Gærplus (50 g/kg feed), composed of the probiotics *Bacillus licheniformis* and *subtilis* and MOS and β-glucans derived from yeast cell wall, to piglets from weaning (~35 day), did not reduce the need for individual piglet antibiotic treatment due to PWD [142]. This synbiotic did not affect morphological traits (such as villi height or crypt depth) in the jejunum in pigs slaughtered 11 days post-weaning. In contrast, Wang et al. [143] reported a significant but small effect of a synbiotic containing 1 × 10^10^ CFU *L. plantarum* and 1.5 g fructoologosaccharides (FOS)/kg feed on the incidence of diarrhoea (from 8.52% in the control to 6.23% in the synbiotic group) when measured for 28 days post-weaning. Guerra-Ordaz et al. [82] tested the impact of *L. plantarum* (2 × 10^10^ CFU/day), 10 g lactulose/kg feed, or a combination of both (synbiotic) in an ETEC F4 challenge study with weaners (weaned 24 day). The challenge promoted a mild-moderate diarrhoea and no effect of the synbiotic was detected. The synbiotic resulted in lower ammonia concentrations in the ileum and colon, higher percentage of butyrate in the colon, lower plasma TNF-α and Pig-MAP levels, and higher villous height, a higher number of goblet cells, and lower numbers of intracellular lymphocytes (in the ileum). However, the two components included in the synbiotic product did not have a synergistic effect: more an additive effect. In another ETEC F4 challenge study with weaners (weaning 17 d of age), Krause et al. [97] investigated a mixture of two probiotic *E. coli* strains, 14% raw potato starch, and their combination (synbiotic treatment). No impact of the synbiotic supplementation on faecal score measured during the first 4 days post-challenge was detected. The colon concentration of SCFA and that of acetic acid in the ileum was higher in the synbiotic group but similar to that measured in the group fed with raw potato alone. The concentration of lactic acid was not affected by the treatments. The numbers of ETEC F4 in the ileum, colon, and faeces were reduced by the synbiotic to the same level as that obtained by the probiotics alone, except in the colon. As in the study by Guerra-Ordaz et al. [82], these results indicate that the synbiotic did not have a synergistic effect, but an additive effect with respect to the two components of the product.

The studies described above, as also seen for other strategies, indicate varying results regarding the impacts on PWD/faecal score, and the impact on parameters related to gut health. The studies that investigated the impact of synbiotics and their components separately in the same study indicate that the synbiotics do not have a synergistic effect, but more an additive one, i.e., adding both components at the same time does not result in a greater effect than adding each component separately.

### 4.4. Postbiotics and Proteobiotics

The relatively new concept of postbiotics is emerging as a tool to promote health, but a clear definition has been lacking. Therefore, The International Scientific Association of Probiotics and Prebiotics (ISAPP) convened experts to review the definition and scope of postbiotics and proposed the definition: “a preparation of inanimate microorganisms and/or their components that confers a health benefit on the host” [144]. Effective postbiotics must contain inactivated microbial cells or cell components, with or without metabolites, that contribute to observed health benefits.

The definition would not include substantially purified metabolites in the absence of cellular biomass. Such purified molecules should instead be named using clear, existing chemical nomenclature; for example, butyric acid or lactic acid. Vaccines, substantially purified components, and products (for example, proteins, peptides, exopolysaccharides, SCFAs, filtrates without cell components, and chemically synthesised compounds), and biological entities such as viruses (including bacteriophages) would not qualify as postbiotics in their own right, although some might be present in postbiotic preparations [144]. Furthermore, according to ISAPP [144], to qualify as a postbiotic, the microbial composition prior to inactivation must be characterised; thus, preparations derived from undefined microorganisms are not included in the definition. On the other hand, a microbial strain or consortium does not have to qualify as a probiotic while living for the inactivated version to be accepted as a postbiotic. The criteria for a preparation to qualify as a postbiotic according to Salminen et al. [144] are shown in Table 4.

Proteobiotics, also a new term, are metabolites from probiotics, i.e., products which do not contain microbial cells [145]. The products studied by Hung et al. [146], Kiarie et al. [147], and Nordeste et al. [148] can be considered proteobiotics. In the study by Kiarie et al. [147], feeding a diet supplemented with *S. cerevisiae* fermentation products to ETEC-F4-challenged weaners did not have an impact on faecal score. Measuring total coliforms and ETEC F4 in ileal mucosa, ileal digesta, and in faeces on days 3 and 7 post-challenge showed a tendency to lower total coliforms in digesta on day 3, but a higher number on day 7, and a lower ETEC F4 number in mucosa on day 7 in the piglets fed proteobiotics [147]. No impact on ileal villi height, crypt depth, or their ratio as compared with the group fed a non-supplemented diet was observed. Addition of the products resulted in a lower abundance of Enterobacteriales and higher bacterial richness and diversity in ileal digesta. A cell-free *Lactobacillus acidophilus* La-5 fermentation product was also tested in an ETEC F4 challenge study with piglets when fed at three supplementation levels [148]. Some positive effects on faecal score were detected, the score being rather low in all groups (the lower, more firm) levels. The animals receiving the fermentation product were less likely to exhibit severe symptoms of illness. Another study with a fermentation product from a lactobacillus strain was that by Thu et al. [149]. Here, three combinations of cell-free fermentation products from different *L. plantarum* strains were investigated in weaned piglets. The faecal score was lower (more firm consistency) in two of the proteobiotic groups compared with a control group. As in the study by Nordeste et al. [148], the score values were low in all treatments, indicating that the incidence of diarrhoea was generally low. All proteobiotic combinations resulted in higher faecal lactobacilli numbers, and one of them in lower Enterobacteria counts, higher SCFA concentration, and higher villi height in the duodenum, as well as improved body weight and gain to feed [149]. Another fermentation product, in this case originating from *Bacillus licheniformis* fermentation, supplied at 1 or 4.4 g/kg feed, was observed to reduce the incidence of diarrhoea in piglets when measured through 28 days post-weaning, but not when considering the first 14 days post-weaning [146]. The bacterial richness and evenness in faeces collected on day 28 post-weaning was lower in the group fed 1g/kg of the fermented product. The first product on the market containing proteobiotic technologies is Nuvio from MicroSintesis Inc. for piglets [145].

In summary, the terms ‘postbiotic’ and ‘proteobiotic’ are relatively new, and there is no consensus in the literature in the ways they are used. An effort has been made to develop a clear definition of postbiotics [144], although that of proteobiotic is less clear. There is still limited literature on feeding these products to piglets, but those presented here on proteobiotics show indications of a beneficial impact regarding PWD. However, the level of diarrhoea was low in all treatments (also the control groups), which makes it difficult to draw strong conclusions on the effect on PWD.

### 4.5. Plants and Plant Extracts

Harbouring a wide range of natural bioactive defence compounds, plants and plant extracts are among the often-investigated and debated alternatives to antibiotics and zinc oxide for preventing PWD in piglets [21,150]. Thus, supplemented to feed as dried plant material, as extracts (e.g., essential oils or tannins), alone or in combination with other compounds, such as organic acids, these botanicals (also denoted phytogenic compounds, phytobiotics) represent a strategy aiming, on the one hand, to have antimicrobial effects, but immune-regulating and antioxidant effects may also be involved [151,152,153]. Notably, benzoic acid is often seen used in blends [154,155,156], and could, in principle, be considered a botanical, because it is a natural constituent of many berries and fruits. However, when used in feed, it is typically categorised as an organic acid.

The handling and dosing of extracted and varyingly purified plant compounds seem to be the most precise and controlled strategies, including the use of nature-identical compounds (NIC), i.e., chemically synthesised counterparts of a natural bioactive compound [157]. However, it should be kept in mind that various interactions between different compounds may affect their mode of action. Therefore, the level and nature of the effects of material from various whole plant parts (e.g., powders of leaves, bulbs, or berries) or more or less defined compound mixtures (cocktails) may differ considerably from that of the individual compounds. This seems to be the case for algae, too, as described later.

This section focuses on the two main types of botanicals being addressed and studied in relation to piglet PWD, namely, essential oils and tannins, where individual studies typically include specific representatives and/or blends of these.

#### 4.5.1. Essential Oils

The term ‘essential oils’ typically signifies hydrophobic composite plant extracts from certain aromatic plants such as various spices, garlic, and chili, comprising, for example, terpenes (carvacrol, thymol, and eugenol), aldehydes (cinnamaldehyde), alkaloids (capsaicin), and sulphurous compounds (allicin). Extracted essential oils often have a few dominating compounds (such as carvacrol and thymol in essential oils of oregano and thyme), but may in total comprise dozens of different compounds; the term is, however, also used for individual compounds, such as carvacrol or thymol [151,158].

The antimicrobial activity of essential oils is partly related to their hydrophobicity, and hence, the capacity to penetrate lipid cell membranes and mediate disruption of the membrane integrity and increase membrane permeability, but essential oils may also affect microbial quorum sensing, for example, resulting in impaired or inhibited toxin production or biofilm formation [159,160].

Cell membranes differ among bacteria with respect to the lipid composition, which affects their sensitivity towards essential oils; in addition, the lipopolysaccharides in the outer membrane of Gram-negative bacteria comprise a hydrophilic barrier against the penetration of hydrophobic compounds [161]. The outer membrane of Gram-negative bacteria is a target for the action of essentials oils, as observed by the release of lipopolysaccharides upon essential oil exposure [162], but Gram characteristics per se do not determine bacterial sensitivity towards essential oils [159,161]. Moreover, and as suggested by Bouyahya et al. [159], because extracts of essential oils typically contain a variety of chemical constituents, it seems probable that more than one mechanism is responsible for their antimicrobial activity. However, in vivo studies with weaning piglets have indicated that the mode of action of essential oils may include the promotion of commensal Gram-positives, e.g., *Lactobacillus* sp., as well as the suppression of potential pathogenic Gram-negatives, particularly *E. coli* [163,164,165].

As for other compounds, such as organic acids, the microencapsulation of essential oils has been investigated with the aim of conferring stability to the bioactive compounds during feed processing and passage through the upper GI-tract, and enable a more or less controlled release along the GI-tract [166,167]. An attempt to protect thymol via gluco-conjugation was not successful, however. Thus, the unprotected form (3.7 mmol/kg feed), and not the protected form of thymol was observed to reduce the faecal score and incidence of diarrhoea of piglet (6 pens × 2 piglets per treatment, weaning age 28 day); neither of the two formulations affected the microbiota composition [152].

Potential synergistic effects of essential oils and organic acids have been studied [167,168], and it has been suggested that essential oils may render the cytoplasmic membrane more penetrable to organic acids, which can then gain easier access into the cytoplasm of the target bacteria [151].

As outlined above, the term ‘essential oils’ is used in the published literature to signify a variety of more or less defined crude extracts as well as individual compounds or mixtures of the latter, complicating direct comparison between studies.

Four non-challenge and six ETEC challenge studies reported effects on piglet PWD (Table 5).

Working with non-challenged piglets, two studies reported significantly positive effects of essential oils on PWD in the immediate post-weaning period [154,170]. Another study [169] observed that essential oils significantly reduced diarrhoea and improved growth performance in a dose-dependent manner; however, they only reported diarrhoea observations for the full trial period (8 weeks). Similarly, another study [155] reported improved piglet growth performance and a tendency for improved faecal score, but again only for the full trial period of 6 weeks.

In challenge setups, working with ETEC F4 or F18, three out of six published studies reported no effects on PWD of the involved essential oil treatments [156,171,173]. Two studies reported essential oils (either as a blend or as individual oils) to reduce diarrhoea in ETEC-F4- or ETEC-F18-challenged piglets [166,174]. Finally, one study reported an essential oil blend to reduced faecal score in ETEC-F4-challenged piglets, but only when fed in combination with an enzyme blend of xylanase and ß-glucanase [172].

In conclusion, based on the referred studies, essential oils have shown potential for reducing piglet PWD development, and there seems to be no doubt that certain essential oils possess specific antimicrobial effects towards ETEC. However, a major challenge is that several studies mix essential oils with a considerable amount of organic acids, in particular benzoic acid, which complicates the evaluation of the essential oil effects per se. Another major challenge for evaluating the results more generally and providing recommendations is clearly that none of the studies are directly comparable, because they differ considerably with respect to the type, amount, purity, and combinations of the involved essential oils.

However, using blends of different plant compounds may be one way of reducing the concentration of the individual compounds, thus avoiding dose-dependent negative effects, as well as potentially obtaining positive synergistic effects. Thus, we have conducted in vitro studies testing the antibacterial effect of various plant materials, alone and in combination, against ETEC fimbriae types F4 and F18, in gastric and intestinal digesta, and demonstrated synergistic effects (Højberg et al., unpublished). We observed that a combination of ramsons and acid berries (lingonberries or red currants) had a more potent antibacterial effect than each plant material separately. The berries contribute by providing organic acids (e.g., benzoic acid) and the ramsons have a high content of allicin, known for its antimicrobial activity [175]. An in vivo piglet trial further showed a clear reduction in the number of *Enterobacteriaceae*, but not of lactic acid bacteria, in faeces and along the GI-tract, when piglets were fed with a standard weaner diet supplemented with 3% ramsons and 3% lingonberries, included as freeze-dried powder of bulbs and berries, compared with the same but non-supplemented diet (Canibe et al., unpublished).

Moreover, a side effect of essential oils is their intrinsic odour and flavour, potentially affecting feed palatability—one way or the other. Negative effects are of course unwanted; however, improved feed palatability upon essential oil amendment has been suggested to prevent the temporary anorexia pattern typically observed for piglets immediately post-weaning [21].

#### 4.5.2. Tannins

Tannins are part of the defence system in a range of plant species and comprise a group of (poly-) phenolic highly complex (e.g., camelliatannins) to more simple (e.g., gallic acid derivatives) molecules, generally classified into condensed (e.g., afzelechins), hydrolysable (e.g., gallotannins), or complex (e.g., camelliatannins) tannins, with the ability to interact with and precipitate macromolecules, such as proteins, polysaccharides, and alkaloids [176]. Overall, the interaction with proteins could be considered, as compared with essential oils, a more broad-spectrum mechanism by impairing the activity of essential microbial housekeeping enzymes. A prerequisite for inhibiting intracellular enzymes is, however, penetration of the bacterial cell membranes, which may be difficult in particular with Gram-negative bacteria such as ETEC. However, other—and more narrow-spectrum—mechanisms of certain tannins are the potential interactions with and binding/blocking of extracellular bacterial protein appendages, such as ETEC F4 and F18 fimbriae, as well as the secreted ETEC LT and ST enterotoxins and/or the respective gut epithelial fimbriae and toxin receptors [176]; overall, there is a range of key factors for the ability of ETEC to induce piglet PWD. Accordingly, tannins may affect ETEC pathogenicity (i.e., incidence of diarrhoea and/or severity) without necessarily affecting digesta ETEC numbers and their faecal shedding per se [177], although concomitant influences on these parameter types may be observed [178]. The variation in published observations probably reflects differences in the type and/or amount of tannins used [176], and emphasises that tannins are a complex group of compounds that should be evaluated individually [179].

Three non-challenge and four ETEC challenge piglet studies reported tannin-mediated effects on PWD (Table 6). In non-challenge setups, two studies reported a reduced PWD for piglets fed hydrolysable tannins as commercial extracts from quebracho and chestnut or chestnut only [174,180]; one of the studies only reported diarrhoea for a full 28-day period post-weaning. A third study reported three levels of pure gallic acid (synthetic) to reduce the incidence of diarrhoea in piglets between day 1 and 10 as well as days 11–21 post-weaning [181]. In challenge setups, two studies reported the use of chestnut extracts (1% and 2% commercial product) to reduce PWD (average faecal score and percentage of piglets with diarrhoea) for ETEC-F4-challenged piglets [177,182]. In two trials, ETEC-F18-challenged piglets were offered tannins (cranberry extract) added to the feed or to the feed and water; they reported reduced ETEC excretion and diarrhoea, but only when administering the tannins in feed and drinking water [183]. Finally, one study offered ETEC-F4-challenged piglets one of three commercial tannin products; reduced ETEC shedding was reported for all three products, but diarrhoea (faecal dry matter percentage) was only reduced for two of the products [178].

Although recent review papers have considered tannins among the more promising antimicrobial plant compounds as alternatives to antibiotics and zinc oxide [150,176,179,185], only a few published in vivo studies have addressed PWD directly. Positive effects are reported, but for some of the ETEC challenge studies, the number of animals included is very low (Table 6). General, as well as more specific antimicrobial effects of tannins are, however, evident from in vitro studies. Similarly to essential oils, tannins may affect feed palatability. In contrast to essential oils, tannins are believed to prevent the ‘unwanted’ animal grazing of plants. Therefore, the palatability issue (bitter taste) needs be taken into account when considering applicable dosing regimens of tannins. Moreover, tannins have historically been considered as anti-nutritional factors for monogastric livestock, but this drawback seems manageable if applied in an appropriate manner, i.e., in sufficiently low concentrations [179].

### 4.6. Algae

Macroalgae, commonly named seaweeds, are macroscopic, multicellular marine algae that are classified into three main groups: brown, red, and green algae. Macroalgae are rich in vitamins, bioavailable minerals, carbohydrates, high-quality proteins [186,187], and n-3 and n-6 polyunsaturated fatty acids [188,189]. Moreover, they possess a broad range of bioactive compounds (e.g., phenolic compounds, carotenoids, tocopherols, polysaccharides, and peptides) with prebiotic, antimicrobial, antioxidant, anti-inflammatory, or immunomodulatory properties [186,190,191,192].

Due to their bioactive properties, macroalgae, and particularly macroalgae-derived compounds such as laminarin (**LAM**) and fucoidan (**FUC**), have attracted considerable interest as potential alternatives to antibiotics and ZnO in preventing PWD in pigs. LAM and FUC are water-soluble polysaccharides derived from brown algae, which escape hydrolysis in the small intestine, passing into the colon. Here, they undergo microbial fermentation [193,194], promoting the growth of beneficial gut microbiota such as lactobacilli [195,196,197] and the production of SCFAs [198]. Findings reported in the reviewed literature are summarised in Table 7.

#### 4.6.1. Laminarin

LAM, an algae-derived β-1,3-glucan, is the main carbohydrate storage in brown algae. Several studies have investigated the health- and growth-promoting effect of dietary LAM in weaned piglets when either supplemented maternally through feeding sows during the gestation and lactation period [196,197,205] and/or as in-feed supplements to weaned pigs [199,200,202,207,209]. Biological mechanisms recurring across studies are associated with prebiotic and antimicrobial activities altering gut microbial composition and microbial fermentation, the modulation of intestinal morphology and barrier function, and immunomodulation.

Dietary supplementation with 300 mg crude or highly enriched LAM/kg feed fed to weaned pigs ad libitum resulted in increased faecal dry matter content and reduced faecal scores 7 to 14 days post-weaning [199,207,209]. Moreover, the dietary inclusion of LAM had the potential to reduce faecal Enterobacteriaceae [200] and *E. coli* counts [207]. This is in agreement with findings by Kadam et al. [212], demonstrating that LAM can inhibit the growth of *E. coli*, *S.* Typhimurium, *Listeria monocytogenes*, and *Staphylococcus aureus* in vitro. Furthermore, dietary LAM has been suggested to possess prebiotic activities when fed post-weaning [199,200] and following maternal supplementation [196,197]. Prebiotic effects were associated with increased colonic and faecal lactobacilli or faecal *Prevotella* counts promoting the production of beneficial SCFAs, particularly butyrate [196,197,199,200]. Despite the positive impact on PWD, studies by Walsh et al. [208] and McDonnell et al. [207] did not identify changes in the lactobacilli population or SCFA production in response to dietary LAM. In addition, Kim et al. [201] reported that the observed reduction in diarrhoea frequency (29.01% vs. 17.28%) of *E.*-*coli*-challenged weaners following the dietary inclusion of an unspecified algae-derived β-1,3-glucan (108 mg/kg) was likely due to enhanced gut integrity (upregulated claudin, occludin, and MUC2 expression in jejunal mucosa). Furthermore, feeding β-glucan enhanced the pig’s immune status by stimulating T-cell activation and reducing inflammation.

In the study by Bouwhuis et al. [197], effects of maternal supplementation with 1.0 g LAM/day from day 109 of gestation until weaning and post-weaning dietary supplementation with 300 mg LAM/kg feed alone or in combination were investigated in weaned pigs challenged with *Salmonella* Typhimurium (1 × 10^8^ CFU). Both maternal and post-weaning supplementation were able to reduce faecal scores. Moreover, maternal LAM supplementation reduced the faecal counts of *S.* Typhimurium by 0.35 log gene copy numbers and *E. coli* counts in the colon and rectum. The post-weaning supplementation of LAM alone did not reveal antibiotic effects either on *S.* Typhimurium or *E. coli*. Instead, pigs offered dietary LAM post-weaning exhibited increased total SCFA and butyric acid and reduced acetic acid concentrations in the colon and rectum. Interestingly, effects on microbial fermentation were not observed in pigs weaned from LAM-supplemented sows and offered a diet supplemented with 300 mg/kg LAM post-weaning. In contrast, in the study by Heim et al. [196], the maternal supplementation of 1 g LAM/d from day 107 of gestation until weaning did not affect colonic Enterobacteriaceae counts in weaned pigs, nor improved faecal scores. Although the incidence of diarrhoea was not affected, maternal LAM supplementation increased the ileal villi height and downregulated pro-inflammatory cytokines (ileal IL-6, colonic IL-8), suggesting that weaned pigs had enhanced gut integrity and gut health [196].

#### 4.6.2. Fucoidan

FUC is a fucose-rich sulphated polysaccharide extracted from the cell wall of brown algae. Proposed biological activities include antimicrobial, immunomodulatory, antioxidant, and antiviral effects [203,209,213,214], and because FUC is a non-digestible polysaccharide, it may also have prebiotic effects [203]. However, studies assessing the effect of dietary FUC on PWD are limited and the findings are inconsistent and less pronounced than those observed for dietary LAM.

In the study by Rattigan et al. [203], dietary supplementation of 250 mg/kg of an FUC-rich extract derived from *Ascophyllum nodosum* improved faecal scores during the early transition period (0–14 days post-weaning) without affecting growth performance. Moreover, propionate, valerate, and total SCFA concentrations in the colon were higher, and butyrate concentrations tended to be higher in pigs supplemented with the FUC-rich extract. SCFAs are involved in the maintenance of colonic homeostasis and promote the absorption of sodium and water [215,216]; therefore, the authors suggested that the improved faecal consistency in FUC-supplemented pigs was likely related to increased water absorption. Interestingly, FUC supplementation had no effect on the colonic microbiota composition, which was in contrast to earlier studies reporting an increase in lactobacilli abundances in the proximal and distal colon [195], as well as in the faeces [207,209] of weaned pigs receiving dietary FUC (240 mg/kg). Despite the increase in lactobacilli abundances, neither Walsh et al. [209] nor McDonell et al. [207] identified positive effects on faecal SCFA concentrations and positive effects on faecal scores were first reported from day 14 to 21 and day 21 to 35 post-weaning, but not during the early transition period [209], or not at all [207].

#### 4.6.3. Laminarin and Fucoidan Interaction

The dietary inclusion of combined LAM and FUC showed positive effects on PWD alleviation when supplemented maternally [205] or post-weaning [207,209], but results were less promising than those observed for supplementation with LAM alone. Despite positive effects on PWD, studies suggest that LAM and FUC have different modes of action with antagonistic effects. For example, in the study by Walsh et al. [208], supplementation with 240 mg FUC/kg feed alone reduced Enterobacteriaceae, but when combined with 300 mg LAM/kg feed, this effect was not observed. Similarly, Lynch et al. [195] observed a reduction in Enterobacteriaceae and an increase in butyric acid in the colon of LAM-supplemented pigs, but again, these effects were negated when LAM and FUC were supplemented together. In the study by McDonnell et al. [207], pigs supplemented with 240 mg/kg FUC had increased lactobacilli populations, but when combined with 300 mg/kg LAM, this benefit was lost.

In line, O’Shea et al. [204] investigated the effect of the interaction between macroalgae-derived extracts (300 mg LAM/kg feed and 240 mg FUC/kg feed extracted from *Laminaria* spp.) and ZnO (3.1 g/kg from 0 to 20 days, 2.5 g/kg from 21 to 40 days post-weaning) on aspects of growth performance, digestibility, and gut health in weaned piglets, concluding that macroalgae-derived extracts and ZnO improve growth performance when given alone, but not when given in combination, and that biological effects of macroalgae-derived extracts on selected digestibility and faecal characteristics were markedly different when compared with that of ZnO. In contrast to studies by Walsh et al. [209] and McDonnell et al. [207], supplementation with macroalgae-derived extracts alone did not reduce faecal scores, but decreased faecal *E. coli* counts by 30-fold. However, the reduction in faecal coliforms was not observed when macroalgae-derived extracts were given in combination with ZnO.

#### 4.6.4. Supplementation of Intact Macroalgae

Dietary supplementation with intact macroalgae as alternatives to antibiotics or medical ZnO in preventing PWD has been less successful [142,211,217] and might reflect potential antagonistic effects between LAM and FUC, as well as other non-digestible polysaccharides, polyphenols, and minerals. In a recent large-scale commercial experiment, Satessa et al. [142] assessed the impact of replacing dietary medical ZnO with alternative products including macroalgae, probiotics, or synbiotics on weaner piglet performance, the incidence of diarrhoea, and gut development. The macroalgae product utilised contained multiple unspecified species of brown, green, and red algae, and a feed analysis revealed the presence of LAM, FUC, algin or alginate, mannitol, fucoxanthin, and rhamnose sulphate as potential bioactive compounds. Supplementation with 15,000 mg/kg of the macroalgae product during the entire weaner period had no effect on piglet health or performance [142]. Previous studies with intact brown macroalgae reported similar results in weaned pigs [211,217] or even reduced performance when fed to finisher pigs [218].

#### 4.6.5. Supplementation of Polysaccharides Derived from Green Macroalgae

Zou et al. [210] found positive results after dietary supplementation with different concentrations (200, 400, and 800 mg/kg feed) of a polysaccharide complex (rhamnose, glucose, glucuronic acid, xylose, and galactose) derived from the green seaweed *Enteromorpha mediates*. Pigs receiving either concentration of the polysaccharide complex showed a reduced incidence of diarrhoea and mortality rates. Moreover, the inclusion of 400 and 800 mg/kg feed improved growth performance and intestinal health, possibly via mechanisms associated with reduced oxidative stress, improved intestinal morphology and barrier function (upregulation of ZO-1, claudin-1 and occludin expression), improved immune status (increased serum IL-6 and TNF-α levels, upregulated IL-6, TNF-α, TLR4, TLR6 and MyD88 expression), and altered microbial fermentation (increased caecal acetic- and butyric acid concentrations). However, further research of green-macroalgae-derived polysaccharides is required to strengthen these findings.

In conclusion, dietary supplementation with macroalgae-derived polysaccharides such as LAM and FUC may be promising alternatives to antibiotic and medical doses of ZnO in reducing the prevalence of PWD. Both LAM and FUC possess prebiotic, antimicrobial, and immunomodulatory properties. Dietary supplementation with either LAM or FUC alone has the potential to beneficially modify intestinal morphology, gut barrier function, immune status, and microbial fermentation. However, effects are lost when offered in combination, suggesting different modes of action. Moreover, the dietary inclusion of LAM alone seems more effective in reducing PWD than the dietary inclusion of FUC alone or in combination. Supplementation with intact macroalgae has been less successful in the immediate post-weaned pig diet, which might reflect potential antagonistic effects between macroalgae-derived compounds.

### 4.7. Antimicrobial Peptides

Antimicrobial peptides (**AMPs**), also known as host defence peptides, are short and generally positively charged peptides found in a wide variety of life forms, from microorganisms to humans. In higher organisms, AMPs constitute important components of innate immunity, protecting the host against infections. In contrast, bacteria produce AMPs (bacteriocins) in order to kill other bacteria competing for the same ecological niche. Many AMPs exhibit a broad range of antimicrobial activity covering both Gram-positive and Gram-negative bacteria, as well as fungi, viruses, and unicellular protozoa. In addition to having a direct antimicrobial activity, several AMPs exhibit an ability to modulate the innate immune responses of the host, and thereby indirectly promote pathogen clearance [47,219,220,221]. AMPs are oligopeptides with a variable composition and number of amino acids, i.e., 6–100 [219,220,222]. By 2013, more than 5000 AMPs had been discovered or synthesised [47], and more than 2000 were natural peptides [219].

Unlike antibiotics, which target specific cellular activities (e.g., the synthesis of DNA, protein, or cell wall), AMPs target the lipopolysaccharide layer of cell membrane, which is ubiquitous in microorganisms [47]. Importantly, this makes it difficult for bacteria to develop resistance against these peptides, i.e., due to their ability to disrupt bacterial membranes via non-specific electrostatic interactions with the membrane lipid components [220,223,224]. Therefore, antimicrobial peptides may have potential as an alternative to antibiotics for use in livestock production. Not less important, the fundamental differences between microbial and mammalian membranes protect mammalian cells against AMPs and enable the selective action of these peptides [47,219,220,221].

The antimicrobial activity of a given AMP is specifically related to its amino acid composition and physico-chemical properties, such as positive net charge, flexibility, size, hydrophobicity, and amphipathicity. Marginal changes in peptide residue sequence are normally followed by major changes in antimicrobial activity [225]. AMPs consist of amino acids; therefore, it is relatively easy to modify the structure. It is possible to make fully synthetic peptides by chemical synthesis or using recombinant expression systems. This gives the opportunity to modify existing AMPs and to design new synthetic AMPs. Such modifications have the potential to change the targets of AMPs and improve the stability of AMPs against proteases [47].

Studies on the impact of AMPs on PWD in pigs are relatively new, with most published studies being conducted during the last decade. Zhang et al. [224] injected chemically synthetised AMP WK3 (2 mg/kg body weight) to ETEC-F4-challenged piglets and measured a lower incidence of diarrhoea, higher growth performance, and higher villus height in the ileum than in the control group (ETEC-challenged) in a six-day-long study (Table 8). WK3 also downregulated the expression of inflammatory factors, i.e., IL-1α, TLR-4, and MyD88, in jejunal mucosa, and increased the oxidative capacity in the jejunum, i.e., reduction in malondialdehyde concentrations and increase in total-superoxide dismutase activity. Cao et al. [226] studied the same AMP (WK3) and provided it to weaners via the feed (50 mg/kg feed) in a 6-day ETEC F4 challenge study. They also reported higher body weight gain and feed intake and lower diarrhoea index (not statistically analysed), as well as lower *Enterobacteriaceae* counts in the caecum lumen, and increased oxidative capacity, i.e., higher GSH-Px in jejunum, and lower IL-1α and TLR-4 expression in jejunum mucosa. The chemically synthetised AMP cathelicidin-BF was intraperitoneally injected to weaners (0.6 mg/kg body weight) during the first 7 days post-weaning, resulting in a decreased diarrhoeal index, increased daily gain and daily feed intake, lower serum concentrations of the pro-inflammatory cytokines IL-6, IL-8, and TNF-α and the anti-inflammatory cytokines IL-10 and TGF-β, and IgG, and higher levels of IgA compared with a control group [227]. Furthermore, piglets fed cathelicidin-BF exhibited lower expression levels of IL-6, IL-8, IL-22, IL-10, and TGF-β in the jejunum and ileum, and higher expression levels of ZO-1, occludin, and claudin-1 in the jejunum and colon. Higher villi were also observed in the small intestine of piglets fed cathelicidin.

Despite positive results, the intraperitoneal administration of AMP to reduce PWD in pig production, as practiced in several of the described studies, would make this mode of administration irrelevant in praxis.

Including a recombinant Microcin J25, a plasmid-encoded AMP isolated from a faecal strain of *E. coli*, at various doses (0.5, 1, or 2 mg/kg feed) in weaner feed, increased weight gain and feed intake, especially in the first two weeks post-weaning, but impaired the gain to feed ratio, reduced the diarrhoea index, reduced serum concentrations of the pro-inflammatory cytokines IL-6, IL-1β, and TNF-α, and increased the serum concentration of the anti-inflammatory IL-10 [223]. The serum concentrations of D-lactate, diamine oxidase, and endotoxin were lower in the piglets fed Microcin J25, indicating a better epithelial barrier function. Furthermore, faecal *E. coli* numbers were reduced and those of Lactobacillus and Bifidobacterium increased by administering the AMP [223]. Providing a chemically synthetised porcine beta-defensin-2 (pBD-2) (5 mL 0.1mg/mL twice daily) orally for three weeks to weaners challenged with *E. coli* K88 resulted in improved daily weight gain, feed intake, and feed conversion [229]. Jejunum villus height and the expression of insulin-like growth factor-I were higher in the pBD-2 group than in the control. The haemolytic *E. coli* scores from rectal swabs and the numbers of *E. coli* in the caecal digesta of the pBD- 2 were lower, as were the gene expression levels of Toll-like receptor 4, TNF-α, IL-1β, and IL-8 in the jejunum mucosa of the pBD-2 compared with the control group. Furthermore, the numbers of lactobacilli and bifidobacteria in the caecal digesta of the pBD-2 group were higher than those of the control. Peng et al. [230] found positive effects when feeding weaners a recombinant pBD-2 (1, 5 or 15 g/kg feed) for 28 d. They measured an increased body weight, average daily weight gain, average daily feed intake after day 14 post-weaning (but not from day 0 to 14 post-weaning), and intestinal villus height in the duodenum and jejunum (day 28 post-weaning), and reduced incidence of PWD (5 or 15 g/kg feed) when measured from day 1 to 28 post-weaning. The authors tested the stability of this AMP at different pH values (2 to 10) and high temperatures (100 °C), and found high values [237], which suggests that pBP-2 can maintain its activity during pelleting of feed and passage through the GI-tract [230]. Compared with a control group, weaners fed the recombinant AMP Plectasin (60 mg/kg feed), a defensin peptide isolated from a fungus, showed a lower feed to gain and diarrhoea rate, higher villi in small intestinal mucosa, increased numbers of Bifidobacterium in the ileum, and higher expression levels of the tight junction proteins CLDN1 and ZO-1 in the small intestinal mucosa of weaners in a three-week study [231]. The recombinant Ceprocin AD, isolated from the silkworm *Hyalophora cecropia*, fed (400 mg/kg feed) to weaners challenged with ETEC K88, numerically reduced the incidence of diarrhoea, and significantly increased weight gain and the gain-to-feed ratio, villus height, and villus-to-crypt ratio in the ileum measured one week after the challenge [221]. The fed piglets Ceprocin AD had also higher levels of secretory IgA in the jejunum and of serum IgA and IgG compared with the control. The results on pro-inflammatory factors IL-1β and IL-6 were not in accordance with the effect seen in other studies, where reductions in pro-inflammatory cytokines are generally reported [223,224,228,229].

Piglets provided a feed supplemented with colicin E1, a bacteriocin produced by *E. coli* (11 or 16.5 mg/kg feed), had a lower incidence and severity of PWD caused by F18-positive ETEC (at the high AMP dose), improved growth performance, and lowered levels of expression of the genes for IL1β and TNF-β in ileal tissue measured four days post-challenge [232].

Wang et al. [233], investigated the impact of lactoferrin on various parameters in weaners, including PWD. Lactoferrin is a glycoprotein synthesised by glandular epithelial cells and secreted into mucosal fluids that bathe the body surface. The highest levels of LF are detected in colostrum and milk, with lower levels detected in tears, nasal fluids, saliva, pancreatic, gastrointestinal, and reproductive tissue secretions. Several biological functions have been described for LF, including the regulation of iron homeostasis, host defence against a broad range of microbial infections, anti-inflammatory activity, regulation of cellular growth, and differentiation and protection against cancer development and metastasis [238]. The addition of lactoferrin to the feed (1g/kg feed) resulted in a lower incidence of diarrhoea when measured for 30 days post-weaning [233].

The impact of feeding with cocktails of AMPs has also been investigated. A study was conducted in five farms where weaners were fed with a mixture of lactoferrin, cecropin, defensin, and plectasin (2 or 3 g/kg feed) for 32 days. Improved average daily gain, average daily feed intake, and gain to feed, and reduced incidence of diarrhoea and mortality were reported when AMPs were included in the diets [234]. In another study, the lactoferrampin-lactoferricin AMP, expressed in the yeast *Pichia pastoris*, was fed to ETEC-challenged weaners for 21 days [235]. Providing the AMP resulted in increased body weight gain and gain-to-feed ratio, numerically decreased diarrhoea rate, and serum IgM and IgG. The serum levels of IL-1β, IL6, and TNF-α were not affected. Feeding an AMP composed of Musca domestica larvae antibacterial peptide and porcine defensin at a ratio of 50:50 to weaners, Shi et al. [236] observed an improved diarrhoea score the first week post-weaning and higher daily gain, but no effect on serum IgG, IgA, and IL-1b, IL-2, IL-6. A reduction in faecal coliforms and increased Lactobacillus and Bifidobacterium was also observed.

As mentioned above, it is easy to change the characteristics of an AMP, even with small modifications. However, predicting the results of these changes is still challenging. Additionally, naturally synthesised AMPs are found only at low levels in their respective host organisms. Therefore, there is a need to increase the efficiency and final yield of peptides during their production to allow their economic exploitation. According to Peng et al. [237], the stability of defensins limits their application as antibiotic treatments [239]. For example, nisin is inactivated under neutral and alkaline conditions [240], whereas AS-48 is deactivated after 5 min at 100 °C [241].

The studies described above show promising results regarding the use of AMPs as alternatives to antibiotics or administering medical doses of ZnO to reduce PWD prevalence in weaners. However, important issues including low specificity, high manufacturing cost, potential toxicity to animal cells, and lack of a robust guideline for rational design are challenges to be taken into account before AMP can be considered in a more practical setup [47]. The ability of the AMPs to pass the stomach and proximal small intestine and reach its site of action without losing activity is also crucial. On the other hand, because the functional characteristics of AMPs depend on their length as well as their amino acid composition and sequence, there likely exists a wide range of potential bioactive peptides that have yet to be identified or characterised [242].

### 4.8. Bacteriophages

Bacteriophages (phages) are viruses that exclusively infect bacteria. They are the most abundant living entities on Earth, with an estimated number of approximately 10^31^, far exceeding other microorganisms [243]. They are common in all natural environments and play an important regulatory role in microbial ecosystems such as the gut microbiome.

Similarly to other viruses, bacteriophages are obligate parasites that entirely depend on the biosynthetic machinery of their bacterial host cell to replicate. Bacteriophages cannot infect just any kind of bacteria, but instead express a strong host specificity, which can range from infecting only one or a few strains of the same bacterial species to a broader host rage of a few closely related species [244].

Depending on their life cycle, bacteriophages are divided into two main types: lytic (virulent) and lysogenic (temperate) phages. Lytic phages destroy (lyse) their host cell shortly after infection (20–40 min to 1–2 h after infection), producing a large number of infectious phage particles capable of infecting adjacent bacterial cells. Hence, lytic bacteriophages will spread as long as there are cells left to infect. In the absence of targeted host cells, the bacteriophages, which cannot survive without a host, will be degraded. In contrast, lysogenic phages do not lyse their bacterial host cell immediately after infection, but instead will integrate phage genetic material into the bacterial genome and co-exist with the bacterial host as prophages. Accordingly, lysogenic bacteriophages will not synthesise large numbers of progeny, but will be replicated as part of the bacterial DNA, which will be transmitted to bacterial daughter cells during cell division.

For therapeutic usage (phage therapy), lytic bacteriophages are favoured due to their inherent antimicrobial property, self-replication ability, and strong host specificity, providing the potential to target specifically pathogenic, but not commensal, bacteria. In contrast, lysogenic phages are often considered unsuitable because they do not lyse the bacterial host cell immediately and may enable the transfer of genetic material such as virulence factors or antibiotic resistance genes from one bacterial cell to another [245]. As natural antibacterial agents, phages are non-toxic [246] and can effectively target both Gram-positive and Gram-negative bacteria. Unlike antibiotics inhibiting a single bacterial physiological process, bacteriophages affect multiple bacterial physiological processes [247], which lowers the risk of developing bacterial resistance. Although bacteria could develop resistance to phages, phages are living organisms that continuously replicate and co-evolve with their bacterial host over time (arms race), and mutation rates of phages exceed those of their bacterial hosts [247]. Another advantage of phage therapy in pathogen control is its ability to self-replicate. Antibiotics must be continually dosed to clear infection; however, phages are able to amplify at the site of infection, which may eliminate the need for applying treatments repeatedly. Novel antibiotic discoveries have stagnated in recent years, whereas phages are highly abundant in nature [243] and can generally be isolated from environments containing the pathogen of interest [244].

Although phage therapy has been used successfully in pigs since the 1920s, it has only recently regained attention as an alternative means of preventing and treating bacterial diseases, improving growth performance and digestibility, and improving food-safety. Only a few studies have investigated the prophylactic and therapeutic efficacy of dietary bacteriophage supplementation to weaning pigs to assess its potential as alternatives to antibiotics and ZnO utilisation in combatting PWD. Results revealed that dietary bacteriophage supplementation not only effectively reduces intestinal pathogen numbers and faecal shed, but also contributes to improved growth performance, gut development, and piglet health around weaning [248,249,250].

In a recent study by Zeng et al. [250], the effect of in-feed bacteriophage supplementation was compared with dietary supplementation with antibiotics. A control diet supplemented with antibiotics (quinocetone and aureomycin) and three treatment diets supplemented either with 200, 400, or 600 mg/kg of a bacteriophage cocktail containing a mixture of phages (10^8^ PFU/g) targeting Salmonella (**S.**), *E. coli*, and *Clostridium perfringens* were fed to 25-day-old pigs, revealing that bacteriophages can be used as an in-feed antibiotic alternative for promoting growth performance and improving piglet health around weaning [250]. Bacteriophage supplementation with 200 mg/kg did not differ from supplementation with antibiotics, whereas increasing phage concentrations to 400 and 600 mg/kg reduced the incidence of diarrhoea, improved gut morphology (reduced CD, elevated VH and VH/CD ratios), enhanced immune capacity (increased sIgA, TGF-α, and serum IL-10, and decreased serum IL-1β and TNF-α) and intestinal barrier integrity (upregulation of tight junction proteins, reduced serum D-lactate concentration and diamine oxidase activity), and had a regulatory effect on gut microbial composition in the caecum compared with the antibiotics diet. In addition to the positive impact on health parameters, bacteriophage supplementation also improved growth performance parameters (final body weight, average daily gain, average daily feed intake, and feed/gain ration), which was in line with other studies demonstrating that bacteriophage supplementation can be used as a growth promoter in growing pigs [251,252].

In line, Hosseindoust et al. [249] showed that dietary supplementation with a 0.10% bacteriophage cocktail (10^9^ PFU/g, targeting *Salmonella*, *Staphylococcus aureus*, *E. coli*, and *Clostridium perfringens*) had beneficial effects on growth performance, digestibility, gut development, and gut microbiota composition in weaned piglets. They demonstrated that the dietary supplementation of bacteriophages resulted in improved faecal scores compared with a basal diet (no dietary supplementation) and a basal diet supplemented with 0.20% organic acids (containing lactic acid, formic acid, and citric acid), and was comparable with results obtained with ZnO supplementation (2500 ppm). Comparable with findings by Zeng et al. [250], supplementation of the phage cocktail resulted in an elevated VH in the duodenum and jejunum, but not in the ileum of piglets. In contrast, duodenal, jejunal, and ileal CD or VH/CD ratios did not differ. Moreover, bacteriophage supplementation had a regulatory effect on the gut microbiota composition, decreasing the concentration of pathogenic bacteria such as *Clostridium* spp. and coliforms, whereas ileal *Bifidobacterium* spp. and *Lactobacillus* spp. were considerably increased. Findings were in line with Kim et al. [252], revealing increased numbers in *Bifidobacterium* spp. and lactobacilli and decreased numbers of *Clostridium* spp. and coliforms following dietary bacteriophage supplementation in pigs. Hosseindoust et al. [249] suggested that bacteriophages included in the cocktail might have been pathogenic to the *E. coli* and *Costridium* spp. Populations, facilitating an intestinal environment favouring the growth commensal and advantageous microbial populations.

In the study by Lee et al. [248], dietary supplementation with a 0.10% bacteriophage cocktail enhanced the intestinal health of weanling piglets, as indicated by an improved faecal score, intestinal morphology, and intestinal absorption. Comparable with Hosseindoust et al. [249], weaned piglets were fed a basal diet without any supplementation (control), and basal diets either supplemented with 2500 ppm ZnO, 0.10% of a bacteriophage cocktail (same cocktail as used by Hosseindoust et al. [249]) or ZnO and bacteriophages combined. The results revealed that both antimicrobial compounds had a beneficial effect on growth performance and gut health [248,249]; however, an interactive effect further enhancing antimicrobial efficacy or growth promotion could not be detected. It was demonstrated that bacteriophage supplementation improved faecal scores more efficiently than ZnO supplementation. In addition, ZnO and bacteriophage supplementation showed comparable effects on gut morphology (increasing VH in the duodenum and jejunum) and growth promotion (increased dry matter and crude protein digestibility, average daily gain, and gain-to-feed ratios) [249]. Moreover, dietary ZnO and bacteriophage supplementation consistently decreased the concentration of *Clostridium* spp. and coliforms, whereas *Bifidiobacteium* spp. and *Lactobacilli* spp. were increased in number.

In addition to its prophylactic efficacy preventing PWD, the utilisation of bacteriophages as a therapeutic treatment against PWD has been experimentally investigated by challenging weaned piglets with ETEC strains. In a study by Jamalludeen et al. [253], the prophylactic and therapeutical potentials of six bacteriophages were investigated in weaned piglets challenged with ETEC-O149. The investigated bacteriophages were selected based on their ability to successfully target and lyse the porcine O149:H10:F4 ETEC strain JG28 in vitro [253]. The six selected bacteriophages were tested either individually or in combination. For prophylaxis, 3-week-old pigs were orally challenged with the O149:H10:F4 ETEC strain JG28 (10^10^ CPF) and bacteriophages were administered orally (by syringe) shortly after challenge (10^10^ PFU), either individually or as a mixture of three phages combined (10^9^ PFU each). For therapeutic use, bacteriophage treatment comprised a mixture of two phages (10^8^ PFU each) administered orally three times at 6 h intervals beginning 24 h after ETEC challenge. The findings revealed that selected bacteriophages were effective in moderating the course of PWD experimentally induced by O149:H10:F4 ETEC when given prophylactically or therapeutically. All six bacteriophages tested individually showed significant prophylactic activity, reducing the mean duration of diarrhoea, mean diarrhoea score, and the shed of O149:H10:F4 ETEC in faeces. The combination of three phages for prophylactic use was able to moderate the course of diarrhoea (reduced duration and severity) but was less effective in reducing the shed of O149:H10:F4 ETEC in faeces. For therapeutic use, a combination of two phages significantly reduced the development of diarrhoea (reduced mean duration of diarrhoea, mean diarrhoea score, and mean composite diarrhoea score) and the number of O149:H10:F4 ETEC bacteria shed in faeces without an apparent reduction in other *E. coli* bacteria present in the gut microbiota.

In recent studies by Han et al. [254] and Lee et al. [255], comparable results were achieved when evaluating the therapeutic efficacy of dietary ETEC-K88-specific bacteriophage supplementation as an alternative therapeutic agent to treat ETEC-K88-induced colibacillosis in weaned piglets. For both studies, comparable challenge models and experimental designs were utilised, and experiments were conducted by the same group. Weaned piglets were challenged orally with either 3.0 × 10^8^ CFU of each of ETEC K88 and K99 ([254]; or 3.0 × 10^10^ CFU of ETEC K88 [255]). In the study by Han et al. [254], bacteriophage supplementation contained a mixture of ETEC-K88- and ETEC-K99-specific bacteriophages (10^9^ PFU/kg), whereas in the study by Lee et al. [255], only the ETEC K88-specific bacteriophage (10^7^ PFU/kg) was supplemented to the diet. In both studies, the same K88-specific bacteriophage was utilised. The results revealed that phage therapy appears to be effective for the treatment of ETEC K88 infection, but not that of K99 infection, in post-weaning pigs. In both studies, the severity of infection symptoms was alleviated by dietary bacteriophage supplementation. Weaners challenged with ETEC K88 and fed a diet supplemented with an ETEC K88-specific bacteriophage exhibited greater weight gain, lower faecal consistency scores, and less faecal shedding and intestinal adhesion of ETEC K88 than challenged pigs fed an un-supplemented diet. Interestingly, faecal shedding and intestinal adhesion of ETEC K99 in the respective ETEC K99 challenge was not reduced by dietary ETEC-K99-specific bacteriophage supplementation [254]. The authors speculated that reduced efficacy of K99-specific bacteriophages may reflect the lower infectivity of the K99 antigen compared with that of K88 in weaners. However, potential confounding effects of ETEC K88 and K99 challenge, as well as the ETEC-K88- and ETEC-K99-specific bacteriophage strains, could not be ruled out [254].

In conclusion, the use of dietary bacteriophage supplementation to prevent and control PWD seems promising. Compared with antimicrobial-free standard control diets, dietary supplementation of bacteriophages enhances gut health and growth performance. Moreover, studies comparing bacteriophage supplementation with a control diet containing antibiotics or ZnO generally reported non-significant differences for gut health, the incidence or severity of diarrhoea, or growth performance, rendering bacteriophages as potential alternatives to antibiotics and ZnO in combatting PWD in pigs. However, those studies were experimental in nature, conducted in controlled environments and under controlled challenge conditions, investigating the effect of specific bacteriophages on a specifically selected infective agent. Moreover, the efficacy of phage therapy is highly influenced by animal age, route of transmission, time, frequency, and concentration of administration, and bacteriophage viability, which can be reduced due to digestive enzymes, extreme pH, high temperatures, host immunity, and other environmental factors. Optimal treatment conditions with respect to PWD thus require further investigation.

Furthermore, it is important to recognise, that the narrow host range currently represent one of the major drawbacks for bacteriophage applications in production settings. Compared with farm conditions, in vivo animal experiments have the luxury of using bacterial isolates for which virulent phages have already been discovered. Under production conditions, however, bacterial isolates may differ from farm to farm, and even between outbreaks. Hence, a priori knowledge of virulent isolate specific phages might not be available, which would require isolate testing before initiating therapy. Phage cocktails, which could empirically offer a broad host range, may be a means of using phages when isolate susceptibility is not yet known, but needs further investigation, especially for therapeutic usage.

### 4.9. Single-Domain Antibodies (Nanobodies)

The immune system of camelids comprises only non-conventional heavy-chain antibody molecules, where antigen binding is mediated solely by one single domain, referred to as VHH. In general, those domains exhibit high specificities for a cognate antigen, high physicochemical stability, as well as a small size, and are thus considered as promising candidates in next-generation biodrugs [256,257]. Compared with conventional antibodies, they are also easier and cheaper to produce [258]. The recombinant antigen-specific, single-domain VHH with dimensions in the nanometre range is also known as a nanobody or single-domain antibody [259].

A few studies have reported the impact of nanobodies in relation to PWD in piglets, which have shown positive results. Virdi et al. [260,261] observed a lower faecal number of ETEC F4 and during fewer days when piglets challenged with ETEC F4 were provided with IgA-like molecules based on the fusion of a fragment crystallizable region with specific VHHs in the feed as compared with a control. Additionally, Fiil et al. [262] studied the impact of a pH-, temperature-, and protease-stable bivalent VHH-based protein, which was shown to bind F4 fimbriae of ETEC in vitro. In an ETEC F4 challenge study, they observed a reduced number of piglets shedding F4 when orally administered the nanobody twice daily for two weeks as compared with a control group. Okello et al. [263] cloned and expressed nanobodies directed against the F4 and F18 fimbriae of *E. coli* on the surface of *L. lactis*. In vitro, the recombinant *L. lactis* strains agglutinated and inhibited the adhesion of cognate F4 or F18 fimbriae expressing *E. coli* to pig villous preparation. In vivo, piglets supplemented with oral anti-F4 L. lactis exhibited reduced faecal *E. coli* shedding after an ETEC F4 challenge. The in vitro inhibition of attachment of F18 fimbriae expressing *E. coli* to piglet enterocytes of nanobodies directed against the lectin domain of the F18 fimbrial adhesin FedF has been demonstrated [258]. Nanobodies as a preventive measure against PWD are under development, but seem promising.

### 4.10. Specific Amino Acids

Nutritional recommendations for amino acids in pigs are mainly based on zootechnical parameters [264] focusing on maximizing protein deposition. However, there is a growing awareness of considering other non-proteinogenic functions of some amino acids [265]. As reviewed by Lopez-Galvez et al. [21], Liao [266], Wang et al. [267], threonine, arginine, glutamine, methionine, and cysteine are important for gut mucosal immunity and barrier function. Glutamine, arginine, threonine, methionine, and cysteine can also assist with relieving the post-weaning stress of piglets by improving gut immunological functions, antioxidant capacity, and/or anti-inflammatory capacity. Glutamine, glutamate, glycine, and cysteine can assist in reconstructing the gut structure after its damage and reverse its dysfunction; lysine, methionine, threonine, and glutamate play key roles in affecting bacteria growth in the lumen [21,266,267].

It has been shown that piglets’ need for tryptophan and threonine increases when the immune system is challenged, as is the case at weaning [268,269] or following a challenge with *E. coli* lipopolysaccharide [270]. Threonine constitutes between 28% and 35% of the amino acids in mucus proteins [271] produced by intestinal cells to protect the underlying mucosa from pathogens such as *E. coli* [272]. Mucins secreted into the lumen of both the small and large intestine are mainly degraded by bacteria in the large intestine [273,274]; therefore, they are not available to the pig [274]. Intestinal mucin secretion represents a significant net loss of threonine, and a constant supply of threonine is necessary to maintain gut function and structure. Zhang et al. [275] (Table 9) supplemented a basal weaner diet with an extra 2 g threonine/kg feed above the National Research Council (**NRC**) recommendations [264], and found improved intestinal mucin synthesis and an increased number of mucin-producing cells, indicating that threonine supply above the requirement may have positive effects on gut health parameters. Similarly, Ren et al. [276] and Koo et al. [277] both reported that the dietary supplementation of threonine above the NRC recommendation improved intestinal morphology parameters.

Tryptophan, in contrast to threonine, is not directly involved in gastrointestinal structure and function. Together with valine, tryptophan is involved in the regulation of appetite and has been associated with the control of stress responses [268]. An important hormone controlling feed intake is ghrelin [278]. Zhang et al. [279] showed that ghrelin mRNA expression in gut tissue and ghrelin levels in plasma were lower in pigs fed a 0.12% vs. a 0.26%/tryptophan diet. Average daily gain (**ADG**), average daily feed intake, and feed efficiency (**FE**) were also improved with the increased intake of tryptophan [279]. Jayaraman et al. [280] also showed that that ADG and FE were improved with an increased tryptophan level in the diet. Capozzalo et al. [281] observed an increase in FE, but only a tendency to increased ADG in weaned pigs infected with *E. coli* and given a basal diet supplemented with extra tryptophan. In addition to being involved in appetite regulation, tryptophan is important in immune responses through increased conversion into kynurenine during inflammation and immune challenges [282]; therefore, piglets may have an increased need for tryptophan during the post-weaning period. In line with this, Jayaraman et al. [280] found that the optimal standardised ileal digestible tryptophan-to-lysine ratio was 4% higher for pigs reared under unsanitary versus sanitary conditions. Additionally, increasing dietary tryptophan has been shown to reduce the intensity of inflammation measured by the acute-phase protein index immediately after infection with *E. coli*, but did not affect the incidence of diarrhoea [281].

In summary, the studies presented here indicate the beneficial gut effects of supplementation with threonine above the nutrient requirement when morphological changes in the gut and stress occur around weaning. Therefore, it is possible that surplus threonine can contribute to prevent PWD, but no studies have been designed to look directly into this. The surplus supplementation of tryptophan relative to the nutrient requirement may also contribute to preventing PWD through the stimulation of feed intake and reduced susceptibility to inflammation, but again, studies on the impact on PWD are needed.

**Table 9 animals-12-02585-t009:** Effects amino acids of algae on various parameters in piglets.

Amino Acid	Dietary Concentrations ^a^	Effect on Productivity and Gut Health Parameters Observed	Study Period	Refs.
Threonine	11.4 g/kg vs. 9.5 g/kg	Improved FE; increased intestinal mucin synthesis and number of goblet cells	From weaning (d 21) until 3 weeks post-weaning	[275,276]
Threonine	11.1, 7.5 vs. 3.7 g/kg	Improved intestinal morphology and mucosa immune function; beneficial effects in maintaining jejunal morphology and integrity and repairing villous damage caused by *E. coli* challenge provided day 16 after weaning	From weaning (day 21) until 18 days post-weaning	[276]
Threonine	0.91 vs. 0.79 % in a simple or complex diet ^b^	Greater benefits with a simple vs. complex diet on intestinal morphology; production of gut microbial metabolites and inflammatory status in the jejunum	From week 1 to 3 post-weaning (day 21.	[277]
Tryptophan	0.26 vs. 0.12 %	Increase ghrelin mRNA expression in the gut and plasma	From weaning (day 28) until 21 days post-weaning	[279]
Tryptophan	0.24 vs. 0.18% SID Trp:Lys	Improved ADG and FE	From week 1 to 3 post-weaning (day 21)	[280]
Tryptophan	0.24 vs. 0.16% SID Trp:Lys	Improved FE, but no effect on ADFI	From weaning (day 21) and until 2 weeks post-weaning	[281]

^a^ Unless specified, each value indicates a supplemental concentration on the top of a basal diet already meeting the dietary requirement for the amino acid. ^b^ The simple diet contained corn, wheat, and soybean meal, whereas the complex diet also contained fish meal, plasma protein, and dried whey, together with the ingredients of the simple diet.

### 4.11. Organic Acids

The addition of ‘acidifiers’ such as citric, fumaric, lactic, propionic, benzoic, and formic acids to weaned pigs’ diets has shown beneficial effects in the pig GI-tract [11], including reductions in the incidence and severity of diarrhoea in the pigs [283]. Studies have tested both mixtures of acids or acids/salts alone and included a number of different SCFA. In our previous review [284], we concluded that organic acids were of special interest for their potential of limiting PWD in pigs; this conclusion was mainly based on the extraction of knowledge from Danish studies, but also some international scientific literature. We presented [284] a total of 35 studies, in which several acids (acidifiers) and mixtures of acids, or mixtures of acids and other additives, had been tested in pigs. However, the impact of organic acids on the treatment against PWD was measured as a secondary response parameter, meaning that the studies were not dimensioned to test this effect. The general more recent international published conclusion [285] is that organic acids and their salts show promising results in controlling intestinal disease and improving the growth performance of pigs, and the authors claimed that consensus results were reported for formic acid (free and salts) in reducing the severity and the duration of diarrhoea on weaning pigs challenged with ETEC F4. As mentioned in the introduction, this strategy will not be further discussed in this review.

### 4.12. Dietary Fatty Acids

The dietary fatty acid (**FA**) composition, and especially the chain length and number of double bonds in the FA, influence the digestion, absorption, and metabolism, as well as the bioactivity of the fat and FA, including inflammatory reactions of the gut. Lauridsen [286] reviewed the effects of dietary FA on gut health and function in pre- and post-weaning pigs, and concluded that during the challenging phase of weaning stress, a strategic use of FA could potentially optimise the growth, function, and health of the pig’s gut. The potential impact of dietary FA on PWD is related to mechanisms involving the intestinal microbiota, the intestinal immune function, and epithelial barrier function, and a significant body of scientific literature has addressed these mechanisms. However, when assessing the impact of dietary fat and FA on diarrhoea in pigs, much less scientific literature is available.

Fatty acids are mainly classified according to the length of the aliphatic tail. Medium-chain fatty acids (**MCFAs**) have aliphatic tails of 6–12 carbons, which constitute the medium-chain triglycerides; long-chain fatty acids (**LCFAs**) have aliphatic tails of 13 to 21 carbons; and very-long-chain fatty acids (**VLCFAs**) have aliphatic tails longer than 22 carbons. Although MCFAs have also been investigated for their role in intestinal inflammation in pigs [198], these FAs have especially been researched due to their antibacterial effects and pH reduction. On the other hand, the main focus on LCFAs and VLCFAs has been directed to their impact on inflammatory immune reactions. Therefore, we first describe studies on MCFAs, followed by those on LCFAs and VLCFAs.

#### 4.12.1. Dietary MCFAs

Several studies [287,288,289,290] have investigated the combination of various forms and blends of MCFAs and SCFAs on the gut health of piglets including measures on diarrhoea. The MCFAs of triglycerides are absorbed directly into the portal blood (for a review, see [286]), and transported to the liver for oxidation, hence providing an immediate energy source for the animal. The MCFAs are also characterised by having a strong antibacterial activity due to their ability to penetrate the semi-permeable membrane of bacteria and damage their internal structures [291]. As an immediate energy source, MCFAs can improve intestinal integrity during inflammatory conditions [292]. Feeding a blend of organic acids and MCFAs (0.2–0.4%) to *E.*-*coli*-challenged piglets around weaning reduced the incidence of diarrhoea [289]. In addition, a recent collaborative study between Chinese scientists and Trouw Nutrition [293], comparing blends of organic acids (SCFAs and MCFAs) to be added to high- and low-digestible diets, showed a reduced incidence of diarrhoea during the first three weeks post-weaning when feeding the experimental blends (various SCFAs and MCFAs). The authors [293] concluded that the results showed a similar growth-promoting effect to that of antibiotics when added to a less digestible diet (i.e., not with high levels of added ZnO). Furthermore, similar blends were tested in a large Danish trial [294], and results in relation to diarrhoea were convincing, i.e., the number of pens treated for diarrhoea was significantly reduced and the blend was as effective as a treatment with 2500 ppm ZnO. However, this study is difficult to interpret in relation to the effect of SCFA+MCFA [294], because the feed contained other components, such as probiotics and fibre, in addition to the FAs, and had a reduced protein content.

Similarly, a study by Long et al. [288] showed that different combinations of SCFAs and MCFAs reduced the incidence of diarrhoea. Han et al. [290] also observed that combinations of SCFAs and MCFAs reduced the rate of diarrhoea in piglets post-weaning. Although the combination of SCFAs and MCFAs seems interesting for the purpose of reducing diarrhoea in pigs, the provision of MCFAs as the only treatment seems less effective. For example, the inclusion of 2% coconut oil (a dietary fat source rich in MCFA) at the expense of soya bean oil had no influence on the incidence of diarrhoea [295]. On the other hand, MCFAs in combination with nanoparticles of ZnO limited the incidence of diarrhoea, although no effect was obtained on faecal consistency and colour [296].

#### 4.12.2. Dietary LCFAs and VLCFAs

The essential fatty acids, linoleic (C18:2n-6) and alpha-linolenic acid (C18:3n-3), cannot be synthesised by pigs (or other mammalian organisms), and must therefore be provided by the diet. These FAs are precursors of the longer chain, higher polyunsaturated fatty acids (PUFAs) of the n-6 and n-3 FA families formed in the tissues by chain elongation and desaturation processes. The origin of the dietary triglyceride fraction (whether it is of vegetable, animal, or marine origin) has a major influence on the dietary fatty acid composition. When taking into consideration that most of the dietary fat fraction (>95%) is in the form of triglycerides, the dietary composition of the fat source offers a potential tool for manipulation of the gut health and function. In relation to human health and immunity, probably the most researched LCFA group is n-6 and n-3 FA [297]; this has also been of major interest in relation to pig nutrition [286]. The conversion of C18:2n-6 and C18:3n-3 to longer-chain PUFAs is limited in pigs (as well as in humans), and C22:6n-3 (docosahexanoic acid, **DHA**) and C20:5n-3 (eicopentanoic acid, **EPA**) must therefore be provided through the diet to be incorporated into tissues and cellular membranes of the body. Fish oil supplementation is an efficient way of providing EPA and DHA, and when provided to pigs after weaning (d 28 of age), DHA and EPA are incorporated in the intestinal mucosal cells measured at day 49 of age [298]. Lowering the proportion of n-6 fatty acids via the incorporation of n-3 LCFA in cellular membranes will reduce the pro-inflammatory eicosanoid synthesis of the n-6 series, which has been found to exert clinical efficacy in human inflammatory diseases [299], and could as such also be relevant for the modulation of intensity and duration of inflammatory immune responses. Marine n-3 FA might also affect the bacterial diversity of colonic microbiota to create a less inflammatory environment for colonic mucosa during human health and disease treatment [299]. However, little is known regarding the effect of n-3 LCFA or fish oil on porcine bacterial composition. Our search for scientific results on the impact of LCFA and fish oil or other VLCFAs revealed few studies addressing the effect of these FAs on diarrhoea in weaned pigs, except for the following: one study in which n-3 PUFA in combination with sodium butyrate and MCFA for gestating sows reduced the incidence of diarrhoea in suckling pigs [300]. Furthermore, a study on maternal supplementation with fish oil showed no influence on the incidence of diarrhoea on the weaned pigs [301], whereas reduced *E. coli* numbers were observed in pigs from another dietary treatment (seaweed extract) in the same animal study. In another study, the diarrhoea rate was significantly higher in piglets provided 6% palm oil in comparison with 6% soy bean oil [302]. Thus, although future recommendations for optimising n-3 and n-6 in diets for pigs post-weaning have been addressed [303], thus far, very little information is available regarding the influence on diarrhoea.

In summary, SCFAs alone or in mixtures with lactic acid are widely added to weaner diets and considered to be a tool to reduce the risk of PWD. Although there seems to be a great potential for dietary LCFA and VLCFA to influence mechanisms of importance for the control of infectious diarrhoea, little evidence has been obtained for the influence on clinical diarrhoea because of limited studies on pigs with this measured parameter. The development of nutritional strategies on dietary FA for reducing the incidence of diarrhoea in pigs could involve blends of SCFA and MCFA, because recent studies have shown potential beneficial effects. However, there are multiple combinations and forms of these compounds, which calls for further investigation to assess their effectiveness.

### 4.13. Milk Replacers and Milk Components

A large and growing body of literature has investigated the potential effect of providing milk replacer and/or other products of milk origin to piglets during suckling and the post-weaning period in order to achieve the optimal development and health of the digestive system. A number of studies have reported the beneficial effects of feeding milk replacer to weaned pigs on the animals’ intestine growth, morphology, and immunity.

Greeff et al. [304] investigated the effect of feeding a nutrient-dense complex milk replacer to piglets prior to weaning. Immediately after weaning (age 21 days), piglets were euthanised, and samples were collected for morphological and functional analyses of the intestinal tract. The results indicated that feeding the milk replacer stimulated intestinal proliferation, leading to increased circular growth of the intestine and increased concentrations of metabolic fermentation products, i.e., SCFA. These findings suggest that the supplementation of nutrient-dense complex milk replacer might enhance the piglets’ robustness and strengthen them through the weaning period by increasing the capacity for the uptake of nutrients [304]. The results obtained by Wolter et al. [305] showed that feeding supplemental milk replacer during lactation produced heavier pigs at weaning, but did not extend to growth performance from weaning to slaughter [305,306]. The results found by Wolter et al. [305] are in agreement with the findings by Azain et al. [307], demonstrating the advantage of milk replacer provision during the suckling period on weaning weight. Zijlstra et al. [308] found that feeding milk replacer the first week after weaning increased feed/energy intake and stimulated pig intestinal development and overall pig performance. Additionally, a case study on farm performed by Pustal et al. [309] showed that the provision of milk replacer during the nursing period contributed to a higher rate of piglets’ survival and resulted in stronger and heavier piglets at weaning.

A number of studies have examined the potential of whey (a by-product of milk processing) in piglet nutrition to enhance the gut immunity and robustness. Thus, Jang et al. [310] investigated the possible effects of whey permeate on growth performance and intestinal health of nursery pigs in a dose–response study. The results indicated that the supplementation of whey permeate could affect the intestinal health of nursery pigs with the potential activation of immune responses and improved enterocyte proliferation. This has a great impact on immune system functionality and on the adequate response to various challenges via immune responses within the intestinal barrier of nursery pigs. The activation of immune responses aids in the prevention of systemic infection and excess inflammation in the post-weaning period [310]. Another study by Thymann et al. [311] investigated the potential effect of a whey-based formula (protein fraction of milk formula based on whey) and a casein+whey-based formula (protein fraction of milk formula based on a combination of casein and whey) on gut function in preterm pigs. The experiment suggested that both whey and casein transiently improved intestinal function. Furthermore, the effect of alpha-lactalbumin (α-Lac)-enriched whey protein on gut and immunity in preterm pigs was investigated by Nielsen et al. [312], who showed that the administration of alpha-lactalbumin (α-Lac)-enriched whey protein improved growth, gut, and immunity parameters in preterm pigs.

To date, several studies have investigated the effect of including porcine natural IgG purified from pig plasma to nursing pigs’ diets. Thus, Jensen et al. [313] investigated the impact of providing bovine colostrum mixed with porcine plasma proteins to newborn pigs. The hypothesis is that porcine IgG would be absorbed more efficiently than bovine IgG. A milk replacer can have a beneficial impact by allowing normal gut growth, but may be inefficient in mediating normal macromolecule transport and disaccharidase activity; thus, by adding porcine natural IgG, piglets receive both immunological protection and a stimulation of intestinal growth and function [313]. The authors concluded that bovine colostrum mixed with porcine plasma proteins may be a useful substitute for porcine colostrum in the artificial rearing of newborn pigs. In an ETEC challenge study with weaners, Hedegaard et al. [314] showed that natural IgG purified from pig plasma and given as a feed supplement significantly reduced shedding of the challenge strain, reduced the proportion of the bacterial family *Enterobacteriaceae*, increased the proportion of families *Enterococcoceae* and *Streptococcaceae*, and generally increased ileal microbiota diversity. Furthermore, the results of Pierce et al. [315] indicated that both porcine and bovine plasma are beneficial to young pig performance during the first week after weaning, and that the IgG fraction of plasma is the component responsible for enhancing the growth rate and feed intake [315,316].

In summary, the studies presented above demonstrate the beneficial effects of providing milk replacer/milk products on parameters defining gut health and gut immunity. However, most studies did not investigate the impact of these effects on preventing PWD, which therefore needs to be investigated.

### 4.14. Dietary Fibre

In the EU, regulation [317] on the provision of food information to consumers defines fibre as ‘carbohydrate polymers with three or more monomeric units, which are neither digested nor absorbed in the human small intestine and belong to the following categories:–Edible carbohydrate polymers naturally occurring in the food as consumed;–Edible carbohydrate polymers which have been obtained from food raw material by physical, enzymatic, or chemical means and which have a beneficial physiological effect demonstrated by generally accepted scientific evidence;–Edible synthetic carbohydrate polymers which have a beneficial physiological effect demonstrated by generally accepted scientific evidence.’

Notably, other components, such as the prebiotic products described above, are also included in this definition, but they are considered and defined separately in this review (inulin, considered a prebiotic, is also included here when used to investigate differences between soluble and insoluble fibre).

According to the regulation [317] on the provision of food information to consumers, the classification of dietary fibre into soluble and insoluble is outdated. This distinction was made on the basis of the different physiological effects of the two types of fibre. However, over the years, considerable scientific research has shown that solubility is not necessarily a determinant of physiological effect. Therefore, in 1998, the FAO/WHO proposed to no longer use this classification [9]. Despite this, the classification of soluble and insoluble fibre (in addition to fermentable and non-fermentable fibre) is still being used in order to group fibre sources and their possible impact on the host. Therefore, in this review, this classification is applied to present the impact of fibre on PWD and related parameters.

Dietary fibre (**DF**) is commonly classified into soluble and insoluble fibre based on the ability to be fully dispersed when mixed with water [318]. Soluble DF typically includes components such as hemi-celluloses (e.g., xyloglucans, galactomannans, mixed-linkage glucans), pectins, gums, and mucilages. On the other hand, lignin, cellulose, and resistant starch are examples of insoluble DF [319]. The positive effects of DF on intestinal function, and thereby, potentially on the PWD susceptibility of piglets, are related to changes in the physico-chemical properties of the digesta, maturation, and integrity of the gut mucosa, changes in microbiota composition, production of SCFAs, and reduction in toxic metabolites from protein microbial fermentation [50].

However, the effect of DF addition to the diet of weaned piglets depends on the type of DF. Insoluble DF primarily acts in the large intestine by increasing faecal bulk and passage rate, modulation of the large intestinal microbiome, and increased production of SCFAs, which may positively affect gut health (Figure 1). Insoluble fibre sources decrease digesta retention time in the proximal GI-tract, which, in turn, may reduce the proliferation of pathogens in the small intestine [50], and thereby, especially relevant to reduce the risk of suffering from PWD. It is also suggested that fibre sources which resemble host receptors might interrupt the adherence of bacteria to the intestinal mucosa, thereby reducing *E. coli* proliferation in the small intestine by blocking *E. coli* adhesion to the gut epithelial cells [48,49,50].

Some soluble DF sources increase digesta viscosity due to their hydration properties, which, on the one hand, is important for effective digestion due to a better accessibility to the substrate by the enzymes, but may, on the other hand, reduce the rate of nutrient absorption in the foregut [21] (Figure 1). Not solubility per se, but the impact of soluble DF on digesta viscosity is considered a key factor in determining the effect of feeding soluble DF to weaners regarding the risk of PWD, and high viscosity promoting *E. coli* growth [320,321,322,323]. Notably, however, in the studies by Hopwood et al. [320], Hoopwood et al. [321], and McDonald et al. [322], the control diets had very low DF levels, which could make it difficult to extrapolate the results of impact of soluble DF to more standard diets.

Hence, in addition to the effects of the functional properties of DF on intestinal physiology and fermentation processes, the selection of specific DF fractions may prevent or stimulate the overgrowth of pathogenic bacteria. There is a need to identify those DF sources that may either increase or reduce the numbers of potential pathogenic bacteria to formulate diets exerting beneficial effects on gut health, thereby preventing PWD [324]. For example, Pascoal et al. [325] observed a higher occurrence of diarrhoea in animals fed diets containing soybean hulls (3%) and citrus pulp (9%), and lower in those fed purified cellulose (1.5%) compared with a control diet.

#### 4.14.1. Insoluble DF

Reductions in the counts of coliform bacteria or the incidence of diarrhoea the first two weeks post-weaning have been described when insoluble DF sources such as oat hulls (2–4%) were added to low-fibre diets, i.e., extruded/cooked rice and maize [326,327], or wheat bran (4%) was included in standard weaner diets [328,329,330]. The particle size of the wheat bran was seen to have an impact, coarsely ground wheat bran being more beneficial regarding PWD and the gut ecosystem, i.e., higher concentrations of SCFAs [330].

In the study by Mateos et al. [327], an interaction between the type of cereal and inclusion of oat hulls in the diet was detected. The inclusion of oat hulls reduced PWD and improved growth performance in weanling pigs fed cooked rice, but had no effect in pigs fed an isonutritive diet based on cooked corn. The content of neutral detergent fibre (NDF, largely corresponding to the insoluble fraction of DF) was lower in the cooked-rice-based diet than in the cooked corn diet, which can partly explain the results. The inclusion of 15% insoluble fibre from wheat straw and oat hull in the diet of weaned pigs the first two weeks post-weaning stimulated the physical adaption of the GI-tract to solid, plant-based feed by increasing the stomach weight and amylase activity in the brush border, and decreased *E. coli* numbers in the ileum and colon compared with a standard cereal-based control diet [331]. Chen et al. [131] investigated the impact of supplementing soluble (1% inulin) or insoluble fibre (1% lignocellulose) alone or in various combinations (total 1%) in diets fed to weaners (24 days of age). They measured improved nutrient digestibility in all diets as compared with the non-supplemented control diet and improved feed-to-gain in the insoluble and 0.5% soluble+0.5% insoluble groups two weeks post-weaning, but observed no effect on diarrhoea. The number of pigs per treatment was low in this study (6 per treatment), however. Similarly, Zhang et al. [332] did not observe an effect of adding 5% corn bran to a weaner diet on diarrhoea the first two weeks post-weaning. After adding 1% or 2% cellulose to a weaner diet, Cho et al. [333] saw no effect on PWD in the first two weeks post-weaning, but identified a higher villus to crypt ratio and lower concentration of plasma COX-2, indicative of a lower inflammation level, on day 14 post-weaning (but not on day 7), which could indicate a healthier status.

On the other hand, in an ETEC F18 challenge study with weaners, Li et al. [334] reported a higher incidence of diarrhoea and increased ETEC shedding in animals fed diets with an added 15% DGGS (insoluble DF source) compared with a control diet, and no effect of adding 10% SBP (soluble DF source). A higher dietary CP (4.2%) in the DGGS diet compared with the control diet was suggested as a possible contributing factor to the obtained results. Berrocoso et al. [335] observed a higher PWD measured during three weeks post-weaning when increasing the DF level of a corn-based weaner diet independently of the type of fibre (2.5% and 5% of sugar beet pulp (SBP), straw, oat hulls, or wheat middlings). Based on their results and those reported by Mateos et al. [327], in which adding oat hulls had a beneficial impact on cooked-rice-based diets but not in cooked-maize-based diets, the latter with higher NDF, Berrocoso et al. [335] argued that additional fibre might be less beneficial in piglets fed high-NDF diets than in piglets fed low-NDF diets. The negative impact was larger for piglets reared under optimal hygiene conditions than those reared under poor hygiene conditions. Montagne et al. [336] found a tendency for a negative impact of increasing DF in a weaner diet (6% SBP+2% soybean hulls) on PWD. However, in contrast to Berrocoso et al. [335], the effect was especially seen in poor sanitary conditions compared with optimal sanitary conditions. When providing piglets additional fibre by adding 2% long-chain arabinoxylans from wheat (lc-AXOS) or 5% purified cellulose or both to a control standard diet, van Hees et al. [337] observed a significant negative effect of lc-AXOS on diarrhoea in the first week post-weaning. Furthermore, intestinal permeability increased when supplying lc-AXOS prior to ETEC (days 7 to 9 post-weaning), whereas after ETEC challenge, cellulose increased permeability. The results, therefore, indicated no beneficial impact of providing either of these two fibre sources (soluble and insoluble) pre-weaning; in contrast, a negative impact was detected.

In contrast to some of the studies just described, not supporting a beneficial impact of adding insoluble fibre to weaner diets, Flis et al. [338], after conducting a review on the impact of fibre substrates, among other parameters, on PWD, concluded that the use of insoluble DF substrates, such as lignocellulose preparations, pure cellulose, cooked or raw oat hulls, and wheat bran, improve faecal consistency and decrease PWD incidence. In comparison with soluble DF sources, insoluble sources improve GI-tract function and health status. Insoluble DF supports GI-tract development, stimulates enzyme activity, and improves gut morphology. They further concluded that due to the various characteristics of fibre used in the pig diets (quantities and solubility), it is difficult to recommend the optimum levels of DF in piglet diets. However, they recommended the addition of 1.5–2.0% of a lignocellulose preparation, 2% of oat hulls, or 4–8% of coarse wheat bran to promote GI-tract development and gut health, and to improve growth performance. In their review, Huting et al. [303] concluded that it is advisable to include moderate levels of inert carbohydrates, i.e., not digested or fermented in the GI-tract, in the diet in order to dilute it and avoid the accumulation of undigested nutrients, reduce the proliferation of pathogens in the small intestine, and to help piglets to increase stomach capacity and restore the activity of brush border enzymes. The authors further speculated that in order to avoid digestive disturbances, the ratio of fermentable to inert carbohydrates during the immediate post-weaning period should be <1.

#### 4.14.2. Soluble DF

Various studies, using low-fibre, semi-synthetic diets based on cooked rice, have shown that soluble fibre sources that increase digesta viscosity, i.e., sodium carboxymethylcellulose and pearl barley, promote *E. coli* growth, and hence, PWD [320,321,322]. A contributing factor could be the fact that an increased viscosity contributes to a longer transit time, enabling the overgrowth of *E. coli*.

Wellock et al. [323] showed that the inclusion of soluble DF (as inulin) as compared with an insoluble fibre source (highly purified cellulose) at either 5% or 15% in diets for weaned piglets challenged with ETEC decreased the occurrence of diarrhoea and improved gut health, as indicated by a lower caecal digesta pH and increased Lactobacillus/coliform ratio when compared with the insoluble fibre diet. An important characteristic of these fibre sources was that none of them affected digesta viscosity. The data indicated that soluble fibre per se, i.e., without increasing digesta viscosity, does not increase the risk of PWD; in fact, it reduces the risk. Supporting these results, Halas et al. [130] conducted an ETEC F4 challenge study with weaners and measured a reduction in PWD incidence and improved faecal consistency when pigs were fed a diet with 8% added inulin as compared with a inulin-free control. As described above, the study by van Hees et al. [337] showed a negative effect of providing piglets with additional fibre by adding 2% long-chain arabinoxylans from wheat (lc-AXOS) to creep feed on diarrhoea the first week post-weaning and on intestinal permeability.

In summary, as the studies described above indicate, the conditions of each experiment can affect the obtained effects of soluble or insoluble fibre sources. Important examples of these factors are: adding a fibre source to a low-fibre diet versus a standard diet; adding purified fibre sources versus fibrous ingredients containing a mixture of fibre structures; and poor sanitary conditions versus optimal sanitary conditions.

Despite the different conditions that can affect the outcome and the lack of consistent results, based on the studies described above and several reviews [38,50,303,335], some strategies regarding dietary fibre inclusion in weaner diets can be proposed. Moderate levels of insoluble fibre sources (e.g., wheat bran, oat hulls, barley hulls, and lignocellulose), preferably in coarse particle sizes, might have positive effects in promoting gut health during the first two weeks after weaning when pigs have a compromised health status. On the other hand, the inclusion of soluble sources, especially resulting in increased luminal viscosity, in the diet for the first two weeks after weaning, particularly with early weaning in farms with poor health status, might be detrimental and promote *E. coli* growth. Once pigs have adapted to solid feed, higher amounts of soluble and fermentable fibre sources can gradually be included in the diet to promote the healthy fermentation of undigested carbohydrates. Soluble fibre sources are easily fermented by the GI-tract microbiota and help to create a stable environment within the GI-tract that can reduce the risk of PWD.

#### 4.14.3. Resistant Starch

Resistant starch (RS), defined as the proportion of starch that cannot be digested by amylases in the small intestine and passes to the colon to be fermented [339], is one of the most potent substrates for increasing large intestinal SCFA production and reducing colonic digesta pH [340,341,342]. Different types of RS exist [343], and physically inaccessible starches (RS1, e.g., intact cereal), resistant granules (RS2, e.g., raw potato starch), and retrograded starch (RS3) are the most relevant RS types applied to pig diets [344].

Feeding newly weaned pigs a diet containing 7% or 14% raw potato starch (RPS) [345] resulted in up to an 88% reduction in PWD the first week post-weaning, but had no effect during weeks 2 and 3 post-weaning. In a study subjecting 17-day-old weaned pigs to a pathogenic *E. coli* (K88 strain) challenge [97], a diarrhoea-reducing effect was also reported when 14% RPS was included in the diet, but only when a mixture of probiotic *E. coli* strains was also provided to the pigs. On the other hand, up to a 20% reduction in PWD between days 8 and 14 post-weaning has been shown [346], when only 0.5% or 1% RPS was included in the diet. Furthermore, the total caecal SCFA concentration was increased, and ileal and caecal digesta pH was reduced after 28 days of RPS inclusion [346]. Supplementing weaned piglets with 5% RPS changed the SCFA profile in the colon towards higher levels of butyrate and increased the expression of genes involved in epithelial barrier function [347], suggesting a beneficial effect on intestinal health. Hedemann et al. [348] also investigated the effect of RPS on the intestinal production of SCFA and morphology in pigs weaned at four weeks of age [348]. Pigs fed 16% RPS had the longest villi, whereas those fed 8% RPS had the deepest crypts. Colon weight and the proportion of butyric acid increased with increasing amounts of dietary RPS [348].

In summary, the large intestinal microbial fermentation in newly weaned piglets can be modulated by the inclusion of RS in the diet and it appears to be an approach to influence the fermentation of carbohydrates instead of dietary protein, and thereby potentially decrease the risk of PWD, because it is known that a healthy gut microbiome plays a role in reducing the incidence of infection and inflammation [349].

### 4.15. Creep Feed

A key factor negatively contributing to piglets’ health in the post-weaning period is considered to be the drop in feed intake or anorexia, most often seen the first days post-weaning; therefore, efforts have been made to stimulate creep feed intake [37,303,350]. The hypothesis behind this strategy is that piglets consuming feed with substrates of plant origin before weaning will eat more after weaning than piglets not consuming any creep feed [41,351,352]. The former will be familiar with the taste and/or consistency of the feed, and the gut will also be more adapted to these feed substrates among other things because the digestive enzymes are more developed (have a higher activity) as a result of creep feed intake. Thus, one of the purposes of feeding creep feed is to ‘train’ the digestive tract to produce enzymes to cope with the post-weaning situation. Creep feed intake will thereby lead to more robust animals and/or heavier piglets, being more resistant to digestive disorders in the post-weaning period, i.e., PWD. The consumption of creep feed can be expected on one hand to reduce the risk of anorexia in the immediate days post-weaning and, on the other hand, to prevent disorders in pigs having a sudden high feed consumption post-weaning because the gut ecosystem, i.e., digestive and microbial parameters, is more adapted to cope with the high load of feed substrates.

However, studies have shown that there is a high variation in creep feed consumption, i.e., not all pigs consume creep feed (35–65%) [38,42], and those consuming creep feed do so at relatively high levels rather late in the suckling period, i.e., the fourth week of life [38,41,353,354]. The amount of creep consumption is considered a key factor regarding the beneficial impact of creep feed on weaning transition; generally, a relatively high consumption is needed in order to have an impact [38]. There are, however, studies indicating positive effects regarding PWD at a relatively low feed intake [355,356]. There are several factors that can affect creep consumption including feed and litter characteristics. The body weight composition of the litter, i.e., uniform or high variation in body weight, influences creep intake of the individuals [353]. Providing the same feed pre-weaning and the initial period post-weaning seems to be advantageous as compared with using different feed formulations pre- and post-weaning [303], or providing creep feed diversity as compared with one type of creep feed [357].

In order to make the weaning transition easier, Oostindjer et al. [358] exposed piglets to an anisic flavour prenatally and/or postnatally through the maternal diet and continuing after weaning. They hypothesized that providing piglets with flavoured food at weaning that matches the flavour in the maternal diet to which they have been exposed to either pre- and/or postnatally, would reduce neophobia and increase the preference for flavour-treated feed, thereby reducing the health (PWD) and welfare problems associated with the weaning process. Postnatal exposure alone did not have an effect. Some impact of prenatal exposure, such as increased food intake and body weight and reduced diarrhoea and some stress-induced behaviour the first few days after weaning, were reported, but the effects were small. In line with these results, Middelkoop et al. [357] concluded that a higher feed diversity, i.e., size, flavour, ingredient composition, smell, texture, colour, etc., rather than flavour novelty alone, seems to affect creep feed consumption to a greater extent.

As another strategy to stimulate creep feed, Kobek-Kjeldager et al. [42] offered liquid feed (feed and water mixed before feeding) from day 12 of life until weaning and continued with the same feed formulation after weaning but fed as dry. Access to supplemental feed in the farrowing pen and the ingestion of supplemental feed did not lead to a higher eating or drinking frequency per piglet on any of the observation days (day 1 and 6 post-weaning) or on the total amount of feed consumed per pen in the first week after weaning. It was observed, however, that access to supplemental feed pre-weaning and piglets eating more the day before weaning shortened the latency to the first eating observation post-weaning.

The impact of creep feed consumption and PWD is not consistent [37,354,355,356,359]. Offering piglets creep feed from day 5 and 10 reduced diarrhoea scores in piglets compared with those offered creep feed from day 15 and those not offered any feed [356]. Increasing creep feed consumption was observed with an increasing creep feeding duration, indicating that the longer duration of creep feeding availability stimulated more pigs to consume more feed. No impact on weight gain pre-weaning or the first week post-weaning was detected, however, probably due to a low creep feed intake. Middelkoop et al. [354] did not identify an effect of offering creep feed from day 2 of life on faecal consistency, feed intake, or growth the first 2 weeks post-weaning. In contrast, Choundry et al. [37], giving piglets access to creep feed from day 3 after birth until weaning, measured a ‘smoother’ relative weight gain (lower coefficient of variation in body weight and a consistent lack of weight loss within piglets of the creep feed group) and a tendency to reach a higher relative weight gain. In addition, the authors observed reduced diarrhoea scores in the first week post-weaning, but not when measured on day 14 after weaning which, according to the authors, indicated an adaptive capacity of young animals [37].

In the study by Carstensen et al. [355], faecal *E. coli* shedding occurred significantly less often and diarrhoea occurrence tended to be lower in piglets that were offered creep feed in the suckling period, but that only showed limited interest in the feed compared with piglets that had more frequent creep feed contact or piglets that had not had access to creep feed at all. It was suggested that intestinal function associated with a voluntary low creep feed contact during the suckling period leads to decreased feed intake just after weaning (two days), and thus reduces the intestinal proliferation of *E. coli* in these piglets. In contrast, Callesen et al. [359] did not identify any difference in PWD the first two weeks post-weaning between creep feed eaters and non-eaters.

Studies such as that by Sulabo et al. [352] show that it is not enough to offer the piglets creep feed if they do not consume it. This was indicated by creep feeding having no effect on pre-weaning or post-weaning growth performance. However, when individual pigs were categorised on the basis of creep feed consumption, eaters had greater gain weight than non-eaters or non-creep-fed pigs. On the other hand, Middelkoop et al. [350] indicated that strategies which do not directly increase creep intake but stimulate the animals to use a feeder could also have a positive impact on the weaning transition. Thus, when offering creep feed, it seems that, as a minimum, piglets should become familiar with the feed or feeder. Familiarisation with a feeder, i.e., the smell, structure of the feed, etc., probably all play a role making this transition easier.

In summary, there only a few studies have investigated the impact of feeding creep feed on the risk of PWD. The results are not consistent, and when positive effects are found, the optimal creep feed intake (high, moderate, or low) is not clear. However, considering the hypothesis that avoiding a period of anorexia or very low feed intake after weaning reduces the risk of digestive disturbances, i.e., PWD, efforts to increase the piglets’ creep feed consumption should be pursued. Increasing the number of piglets within the litter consuming creep feed is also desirable, which seems to be affected by factors varying from feed composition and appearance to litter composition, i.e., the number and condition of the piglets. Furthermore, to correctly interpret the results of these type of studies, it is not enough to offer the creep feed, but the actual consumption should also be registered.

### 4.16. Vaccines

Immunisation by vaccines is common in veterinary practice, and efficient intramuscular vaccines for sows against ETEC are available. These vaccines are used to immunise pregnant sows and thereby stimulate the production of maternal antibodies to protect suckling pigs against ETEC diarrhoea. However, these passively acquired antibodies are rapidly lost at weaning, and therefore do not protect pigs against PWD [360]. ETEC infections are non-invasive; therefore, mucosal immunity rather than systemic immunity is the key factor when attempting to prevent PWD with vaccines. Therefore, for vaccines to be efficient, they need to elicit an active intestinal mucosal immune response with the production of antigen-specific secretory IgA (sIgA) [361].

#### 4.16.1. Live Oral Vaccines for Pigs

Live oral vaccines containing *E. coli* strains have the potential to elicit an active intestinaliga mucosal immune response against ETEC. Currently, one live oral vaccine for pigs against PWD (Coliprotec^®^ F4/F18) is commercially available. The vaccine contains two non-pathogenic (non-toxigenic) *E. coli* strains: *E. coli* O8:K87 (F4ac) and *E. coli* O141:K94 (F18ac) as antigens. The vaccine is orally administered by drench or via the drinking water system to pigs of a minimum age of 18 days. Antibiotics active against *E. coli* cannot be used around the time of vaccination, because these will kill the strains in the vaccine. For vaccine strains to be able to bind to the intestinal mucosa, and thereby elicit its effect, vaccinated pigs need to express intestinal receptors for F4/F18 fimbriae [362]. The claim of the vaccine is that it reduces the incidence of moderate to severe post-weaning *E. coli* diarrhoea in infected pigs and reduces the faecal shedding of ETEC-F4-positive and ETEC-F18-positive in infected pigs. Studies suggest that the vaccine induces cross-protection against F18ab-positive as well as F4ab- and F4ad-positive *E. coli* strains (European Medicines Agency). According to the claim, the onset of immunity is seven days after vaccination and duration of immunity is 21 days after vaccination. In the study by Nadeau et al. [362], pigs were vaccinated with Coliprotec^®^ F4/F18 one day post-weaning and challenged on day seven and twenty-one post-vaccination with either an F4-positive strain (positive for STa, STb, LT, East-1, and F4ac) or an F18-positive strain (positive for STb, LT, East-1, Stx2e, and F18ab) (Table 10). Compared with unvaccinated controls, the vaccinated pigs had fewer days with diarrhoea and less severe diarrhoea in the 4–7 days post-challenge. In another experiment, pigs were vaccinated with Coliprotec^®^ F4 one day post-weaning and challenged on day three, seven, or twenty-one post-vaccination with an F4-positive strain (positive for STa, STb, LT, and East-1) [363]. This study showed that the incidence of diarrhoea was significantly lower in vaccinated pigs compared with unvaccinated controls following challenge at 7 and 21 days post-vaccination. Following challenge day 3 after vaccination, no statistically significant reduction in the incidence of diarrhoea was seen, but the duration of diarrhoea was lower in the vaccinated group.

The majority of ETEC in submissions from cases of diarrhoea and oedema disease in pigs in Denmark are either F4- or F18-positive. However, 20% of the haemolytic *E. coli* isolates tested in 2018 were neither of these. In these cases, the vaccine cannot be expected to have a significant effect. A recent Danish study genomically characterised 83 F4/F18 *E. coli* isolates detected in 64 Danish herds during 2018–2019. F4ac and F18ac were the only subtypes detected [364]. Thus, it seems that the antigens in the Coliprotec vaccine targets the isolates circulating in Danish stables, although the sampling method was not suited to conclude on the overall Danish occurrence of subtypes.

In a survey conducted in 2021, 32 veterinary practitioners (response rate: 18.5%) shared their experiences and views on Coliprotec^®^ F4/F18 [365]. Approximately one-third of the responding veterinarians had no personal experience with the vaccine. In general, evaluations of the effect of the vaccine have varied. The veterinarians pointed out that the multifactorial nature of PWD with feeding and management factors having a large influence seem to explain the lack of effect. The main reason for not using the vaccine and for ceasing use was a lack of effect and price. Notably, the price of an antibiotic treatment against *E. coli* was one-third of the price of the vaccination.

#### 4.16.2. Live Oral Vaccines for Sows

Oral vaccination of pregnant sows with a live attenuated *Salmonella* Typhimurium mutant with a plasmid containing five *E. coli* adhesin genes (F4ab, F4ac, F5, F6, and F41) showed promising results in a challenge trial after inoculation with ETEC at the point of weaning (21 days of age) [366]. Oral vaccines have the advantage of stimulating the gut-associated lymphoid tissues (GALTs) of sows, thereby inducing the production of sIgA, which is transferred to the piglets via colostrum. The vaccine candidate induced a significantly higher IgG level in piglets at one and three weeks of age compared with a traditional intramuscular sow vaccine. However, the study did not evaluate whether the protective effect of the vaccine candidate lasted for longer than 21 days.

#### 4.16.3. Oral Subunit Vaccines for Pigs

Subunit vaccines comprise components of a pathogen, rather than the entire pathogen, able to induce protective immunity [367]. Purified F4 fimbriae have been shown to act as efficient oral immunogens and to elicit IgA and IgG responses comparable with those seen upon oral challenge with live F4-positive *E. coli* [368]. Using F18 fimbriae as immunogens is more complicated. Thus, experimentally, high doses of purified F18 fimbriae did not elicit a protective immune response [369]. However, the application of a vaccine containing the F18 fimbrial adhesin FedF fused with maltose-binding protein and conjugated with F4 fimbriae reduced the faecal excretion of F18-positive *E. coli* upon challenge [370]. In another study, porous tablets were loaded with F4 fimbriae and tested for efficacy for the oral vaccination of piglets against F4-positive ETEC [371]. The study reported on serum antibody responses after immunisation and the faecal excretion of F4-positive *E. coli* after challenge. The authors concluded that “F4 fimbriae loaded porous tablets could be a novel oral vaccination candidate to induce mucosal and systemic immunity against ETEC infections”. Genetically modified tobacco plants expressing high levels of an engineered variant of the major subunit FaeG from ETEC F4 fimbriae may constitute a potential oral vaccine candidate [372]. Thus, chloroplast-expressed rFaeG_ntd/dsc_ (recombinant variant of the FaeG protein in F4) displayed in vitro binding to F4-specific epithelial receptors and inhibited ETEC adhesion to porcine small intestinal villi. However, testing of this protein in in vivo animal models has not been performed.

Another research group has worked with rice-based oral subunit vaccines designed to induce heat-labile toxin (**LT**)-specific antibodies [373]. The idea behind immunisation against LT rather than against adhesins is that despite the variation in types of ETEC fimbriae, the LT produced by ETEC does not vary. Therefore, the induction of toxin-specific neutralisation antibodies rather than adhesin-specific antibodies could be an important strategy, because these vaccines would cover a broader range of pathogenic *E. coli*.

#### 4.16.4. Intranasal Subunit Vaccines for Pigs

In an experimental study, the protective efficacy of an intranasally applied subunit vaccine for F4 ETEC in weanling-age piglets was tested [374]. The study also analysed antibody responses to vaccine antigens F4 and LT. The study concluded that the intranasal vaccine was protective against experimental challenge with F4-positive *E. coli*.

#### 4.16.5. Parenteral Vaccines for Pigs

Parenteral vaccines tend to stimulate the systemic rather than the mucosal immune system, and therefore, are generally not considered to be effective against non-invasive ETEC infections [361]. Interestingly, however, Ruan et al. [360] showed that an intramuscularly applied tripartite fusion protein resulted in the induction of IgA antibodies in both serum and faeces [360]. This study also showed an inhibition of adhesion of F4 and F18 upon challenge and protection against clinical signs after challenge with F4-positive *E. coli*. A clinical effect after challenge with F18-positive *E. coli* could not be demonstrated because the study was unable to experimentally reproduce F18 ETEC diarrhoea in weaned pigs.

Systemic induction of a serum IgA response will result in the secretion of sIgA into the intestinal tract during the systemic response, but this is not necessarily associated with the presence of sIgA antibody-secreting cells or with the induction of a mucosal memory response in the GALT [361]. Experiments have indicated that adding vitamin D3 to intramuscular vaccines with F4 fimbriae can modulate the systemic response towards an intestinal mucosal IgA response. Thus, adding vitamin D3 to intramuscular vaccines resulted in a reduced faecal excretion of F4-positive *E. coli* upon challenge [375].

Multivalent vaccine candidates targeting multiple immunogenic epitopes in ETEC are under development. Thus far, this technology, ‘Multiepitope fusion antigen’ (MEFA) [376,377], has only been tested experimentally with mice as models. These vaccines are designed to mimic the antigenicity of multiple *E. coli* types at the same time and therefore of great potential interest in practical situations.

In summary, because ETEC infections are non-invasive, the active induction of mucosal immunity is very important for achieving protection against PWD. Coliprotec^®^ F4/F18 has shown to be an effective live oral vaccine in cases of PWD caused by predominant types of ETEC in Danish cases, which are also common in many other countries. There seems, however, to be limitations for the practical use, such as the price compared with antibiotics and the multifactorial nature of PWD—indicating that factors other than the adherence of ETEC bacteria to the intestinal wall are important in relation to protection against PWD.

Experimentally, different vaccine candidates have been and are currently under investigation. Multiepitope fusion antigen parenteral vaccines seem interesting, due to their potential ability to protect against a range of ETEC, and not only F4/F18-positive strains. For parenteral vaccines to have an adequate effect on PWD, however, further studies looking further into adjuvants with the ability to modulate the systemic response towards an intestinal mucosal IgA response seem relevant.

Oral vaccines for sows, oral subunit vaccines for pigs, and intranasal subunit vaccines for pigs all need further research assessing whether they will be useful in practice. Generally, the latter two seem to be the most promising, because they are directed at the intestinal mucosa and thereby should have the potential to induce an effective mucosal immune response.

**Table 10 animals-12-02585-t010:** Effects of vaccines on various parameters in piglets ^a^.

Vaccine Type	Vaccination Protocol	Number of Animals	ETEC Challenge, Timing	ETEC Challenge, Dose (CFU)	Challenge to Euthanasia Interval	Endpoints ^b^					Refs.
						FS	GP	FT/D	SIR	IIR	
Live, oral for pigs (Coliprotec F4/F18^®^)	One time one day post-weaning	44 vaccinated + 44 non-vaccinated	7 and 21 days postvaccination	Between 1.3 × 10^9^ and 5 × 10^10^	4–7 days	+	+	+	+	nm	[362]
Subunit, oral for pigs	Three times: 14, 31, and 55 days post-weaning	3 vaccinated + 9 non-vaccinated	7 days after last vaccination	10^11^	Euthanasia before challenge	+	nm	nm	+	+	[370]
Subunit, oral for pigs	Six times: 7,8,9,21,22 and 23 days post-weaning	6 vaccinated + 6 unvaccinated	12 days after last vaccination	2 × 10^9^	12 days post-challenge	+	nm	+	+	nm	[371]
Subunit, oral for pigs	Four times: 60, 74, 88 and 102 days of age	5 vaccinated + 4 non-vaccinated	NS	10^6^ as in vivo intestinal loop challenge	18 h after surgery	nm	nm	+	+	+	[373]
Subunit, intranasal for pigs	Two times: 10 and 17 days of age (gnotobiotic pigs)	13 vaccinated + 4 non-vaccinated	7 days after last vaccination		4 days post-challenge	nm	+	+	+ (tested in other group of pigs)	nm	[374]
Subunit, intramuscular for pigs	Two times: 5 and 19 days of age (gnotobiotic pigs)	3 vaccinated + 3 non-vaccinated	14 days after last vaccination	3 × 10^9^	2 days post-challenge	nm	nm	nm	+	+	[360]
Subunit, intramuscular for pigs	Two times: 8 and 25 days post-weaning	10 vaccinated + 5 non-vaccinated	8 days after last vaccination	10^10^	15 days post-challenge	+		+	+	nm	[375]
Live, oral for sows ^c^	Two times: Week 8 and 11 of pregnancy	6 vaccinated sows + 6 non-vaccinated sows	Piglets of one and three weeks of age	10^9^		nm	nm	+	+	nm	[378]

^a^ Table limited to challenge studies involving pigs. +: A positive effect; nm: not measured. ^b^ FS: faecal shedding of challenge strain; GP: growth performance; FT/D: faecal traits/reduction in diarrhoea episodes; SIR: serum immune response; IIR: intestinal immune. ^c^ Results given for pigs (not sows).

## 5. Conclusions

Many strategies have been explored and investigated during the last few decades to reduce the risk of PWD in pig production. Initially, the focus was to find strategies to reduce the use of antibiotics, and in more recent years, to replace the use of medical ZnO during the first two weeks post-weaning. Although many studies show positive effects, the results are seldom consistent. The issue of bias in published studies, i.e., the tendency to publish positive results, but less so negative results, is always relevant to consider but difficult to cope with, specifically when trying to draw conclusions on the effectiveness of a strategy/additive, etc.

The reasons for the equivocal results observed can be multiple, including factors related to: (i) feed, i.e., ingredients of the basal feed used and form; (ii) experimental treatment, i.e., concentration/level of the treatment or product applied, applied alone or in combination, and duration of application; (iii) animals, i.e., breed, age, and health status; (iv) management, i.e., level of stress, hygiene level, and housing; (v) study design, i.e., number of replicates, length of the treatment applied, and sampling and/or registrations; and (vi) study model, i.e., challenge study, non-challenge study, etc. Furthermore, Wegh et al. [379] indicated that not only differences in effects can be found between specific postbiotics, or between individuals, but also temporal changes in gut microbiota composition could influence the response to interventions. This can probably be extended to most strategies.

It is not possible to rank the strategies described in this review. On the other hand, there seems to be agreement that various initiatives should be applied at the same time (as opposite to only one strategy) and be tailored for the specific production site. The mode(s) of action of the various ingredients or additives are most often different and might, in some cases, even have antagonistic effects, e.g., algae compounds, but several of them have also common mode(s) of action resulting in additive or synergistic effects, e.g., plant materials. More detailed knowledge on the mode(s) of action of the individual strategies would aid in designing the appropriate combinations to be applied.

As concluded in previous reviews and studies, it seems that, to date, no single strategy has proven to be totally effective, and it is probable that the most successful approach on a particular site will involve a combination of more than one feed intervention together with other preventive measures.

Strategies such as the reduction in dietary protein levels or the addition of organic acids are considered to have a positive impact but, as mentioned in the Introduction, were either not dealt with at all or only very briefly here. These are already used, i.e., in several countries where medical ZnO is still added to the diets; therefore, it can be speculated what their effect will be when medical ZnO is no longer used in pig production. They are expected to be beneficial, but the magnitude of this effect can be different.

## 6. Perspectives

In order to document the impact of feeding and other strategies on PWD, studies with appropriate designs for monitoring diarrhoea should be conducted. Additionally, defining and measuring diarrhoea can be challenging and the relevance of changes in faecal score when they are far from a diarrhoeic state should be addressed.

Identifying and agreeing on valid and robust biomarkers/surrogate parameters to measure the efficacy of interventions to reduce the risk of PWD would be of significant help in this research area of trying to prevent PWD.

When investigating PWD, primarily occurring during the first two weeks post-weaning (and not diarrhoea developing in later weeks and due to other pathogens and factors), it might not be relevant to obtain samples and registrations from pigs euthanised at 4–5 weeks, which is often seen in studies investigating PWD aspects. An example of this is the type of fibre, insoluble versus soluble, regarding timing, as discussed above. Therefore, more precision in what being investigated, and therefore, when to measure it, should be pursued.

In general, for the strategies to have a positive and consistent effect, a more tailored approach should most probably be designed for the specific conditions of the production system at the specific time point, as exemplified by the description of the bacteriophages and vaccine strategies. This adds a challenge because much more detailed knowledge of the pathogens present at the specific time point are needed, as well as more detailed information on the animals.

It can be speculated that new production systems and technologies in future pig production will also influence the outcome of the treatments applied, but at the same time provide new possibilities. It is predicted that fewer and larger production units will form the future of pig production, which can make it more feasible to obtain such information. This could represent possibilities to perform diagnoses and design targeted strategies accordingly, at least for a period of time, which increases the overall chance of obtaining the desired beneficial results.

## Figures and Tables

**Figure 1 animals-12-02585-f001:**
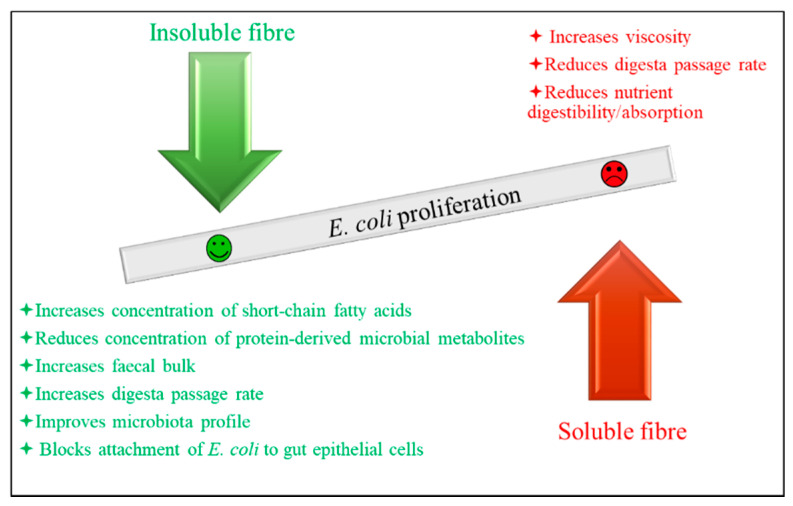
Proposed mode of action behind the effect of insoluble and soluble fibre on *E. coli* growth in the gut of weaners.

**Table 1 animals-12-02585-t001:** Proposed modes of action of various strategies to prevent post-weaning diarrhoea in piglets.

Establisment of a Robust Gut Microbiota	Promoting Maturation of the Gut before Weaning	Inhibition/Reduction of the Growth of Pathogens	Promotion of the Growth of Beneficial Bacteria	Modulation of the Immune Response/Provision of Immune Protection
– Probiotics – Prebiotics – Synbiotics	– Creep feed – Fibre – Milk replacers	– Organic acids – Plant components – Antimicrobial peptides – Bacteriophages – Probiotics – Fibre – Single-domain antibodies	– Probiotics – Prebiotics – Synbiotics	– Milk replacers – Probiotics – Immunoglobulins – Vaccines – Antimicrobial peptides – Plants/plant extracts – Algaes – Amino acids

**Table 3 animals-12-02585-t003:** Effects ^a^ of prebiotics on various parameters in piglets.

Prebiotic	Dose	No of Replicates	Pathogen Challenge	Endpoints ^b^						Refs.
				GP	ND	FT/D	GA	IS	GM	
Chito-oligosaccharides	0.01, 0.02 and 0.04%	10 piglets	None	+ (0.01 and 0.02%)	+ (0.01 and 0.02%)	+	+ (0.01 and 0.02%)	nm	+	[127]
Lactulose	0.5 and 1%	7 pens; 5 piglets/pen	None	+	+	NS	nm	nm	+	[128]
Lactulose	1%	6 pens; 3 piglets/pen	ETEC F4	+	nm	NS	+	+	+	[82]
Chicory	30%	13–14 piglets	ETEC F18	(-)	nm	NS	nm	nm	+	[129]
Inulin	4 and 8%	16 piglets	None	NS	NS	NS	NS	nm	-	[130]
Inulin	1%	6 piglets		NS	+	NS	nm	nm	nm	[131]
β-glucans, glucomannans, MOS ^c^	0.1, 0.2, and 0.3%	6 pens; 4 piglets/pen	None	nm	nm	NS	nm	nm	nm	[132]
Isomalto-oligosaccharide	0.6%	6 pens; 6 piglets/pen	None	NS	NS	+	nm	+	nm	[133]
FOS ^d^ and MOS	0.1% and 0.5%+0.5%	3 pens/10 piglets	None	+	+	+	nm	NS	nm	[134]
MOS	0.2%	8 pens; 4 piglets/pen	None	+	nm	+ only day 7	NS	NS	+	[135]

^a^ Effects. +: positive effect; -: negative effect; NS: non-significant effect; nm: not measured. ^b^ Endpoints. GP: growth performance; ND: digestibility; FT/D: faecal trait/diarrhoea; GA: gut architecture/histology; IS: immune system; GM: gut microbiology. ^c^ Mannan-oligosaccharides. ^d^ Fructo-oligosaccharides.

**Table 4 animals-12-02585-t004:** Criteria for a preparation to qualify as a postbiotic Salminem et al. [144].

Criteria for a Preparation to Qualify as a Postbiotic
▪ Molecular characterization of the progenitor microorganisms (for example, fully annotated genome sequence) to enable accurate identification and screen for potential genes of safetyconcern. ▪ Detailed description of the inactivation procedure and the matrix. ▪ Confirmation that inactivation has occurred. ▪ Evidence of a health benefit in the host from a controlled, high- quality trial. ▪ Detailed description of the composition of the postbiotic preparation. ▪ Assessment of safety of the postbiotic preparation in the target host for the intended use.

**Table 5 animals-12-02585-t005:** Effects ^a^ of essential oils on various parameters in piglets.

Essential Oils	Dose(s)	Replicates	Pathogen Challenge	Endpoints ^b^					Refs.
				GP	FT/D	GMo	GI	GM	
Carvacrol and thymol (mix)	0.03, 0.06, 0.10%	4 pens of 4 piglets	None	+	+ full period only	nm	nm	nm	[169]
Thymol, 2-methoxyphenol, eugenol, piperine, and curcumin (mix with 90% benzoic acid)	0.30%	10 pens of 4 piglets	None	+	+ within day 0–42	NS	Various serum levels	+ (cecum)	[154]
Eugenol, thymol, and piperine (mix with 90% benzoic acid)	0.30%	9 pens of 3 piglets	None	+	(+) tendency	+	nm	+	[155]
Cinnamaldehyde and thymol (mix, microencapsulated)	0.01%	6 pens of 5 piglets	None	+ Day 15–28 only	+	+	Antioxidative capacity (blood, mucosa, liver)	nm	[170]
Cinnamaldehyde, eugenol, carvacrol, thymol, and diallyl disulfide (mix)	0.10%	8 pens of 3 piglets	ETEC F4 day 8 PW	NS	NS	NS	NS	+ >LAB:coliforms	[171]
Thymol, 2-methoxyphenol, eugenol, piperine, and curcumin (mix with 96% benzoic acid)	0.2, 0.3, 0.4%	9 pens of 5 piglets	ETEC F4 day 7–8 PW	+ For some doses and periods only	NS	NS	nm	NS	[156]
Thymol, vanillin, and eugenol (mix with organic acids (fumarate, citrate, malate, and sorbate), microencapsulated)	0.20%	6 pens of 1 piglet	ETEC F4 day 7 PW	NS	+	(+)	NS	NS	[166]
Thymol and cinnamaldehyde (mix +/− feed enzymes)	0.01%	6 pens of 8 piglets	ETEC F4 day 8 PW	nm	+ with enzymes only	+	nm	+ <coliforms	[172]
Cinnamaldehyde (commercial mix with organic acids and permeabilizing complex)	0.15%	6 pens of 4 piglets	ETEC F4 day 4, 5, 6 PW	NS	NS	nm	nm	nm	[173]
Capsicum oeloresin, garlic botanical, and turmeric oeloresin (tested individually)	10 ppm	8 pens of 1 piglet	ETEC F18 day 4 PW	NS	+	+	+	nm	[174]

^a^ Effects +: positive effect; −: negative effect; NS: non-significant effect; nm: not measured. ^b^ Endpoints. GP: growth performance; FT/D: faecal trait/diarrhoea; GMo: gut morphology; GI: gut immunology; GM: gut microbiology.

**Table 6 animals-12-02585-t006:** Effects ^a^ of tannins on various parameters in piglets.

Tannin Source	Dose(s)	Replicates	Pathogen Challenge	Endpoints ^b^					Refs.
				GP	FT/D	GMo	GI	GM	
Quebracho and chestnut (mix with humic acids)	0.75%	6 pens of 14 piglets	None	nm	+	nm	nm	+ >LAB:coliform >Prevotella	[180]
Chestnut (compare to ZnO)	0.10%	6 pens of 6 piglets	None	NS	+ day 1-14	nm	+	nm	[184]
Gallic acid (NB! ZnO in diets)	0.01, 0.02, 0.04%	6 pens of 4 piglets	None (NB! LPS)	NS	+ both periods	+	+	nm	[181]
Chestnut	1.0, 2.0%	9 pens of 2 piglets	ETEC F4 day 4 PW	+ (for 2% only)	+	nm	nm	nm	[177,182]
Cranberry	1.0%	4 pens of 1 piglet	ETEC F18 day 8 PW	NS	NS	nm	nm	nm	[183]
Cranberry	1.0% + 0.1% in water	3 pens of 1 piglet	ETEC F18 day 8 PW	NS	+	nm	nm	nm	[183]
Three commercial tannin products (tested individually)	1.0%	1 pen of 5 piglets	ETEC F4 day 6-7 PW	NS	+ 2 of 3 products	nm	nm	nm	[178]

^a^ Effects. +: positive effect; −: negative effect; NS: non-significant effect; nm: not measured. ^b^ Endpoints. GP: growth performance; FT/D: faecal trait/diarrhoea; GMo: gut morphology; GI: gut immunology; GM: gut microbiology.

**Table 7 animals-12-02585-t007:** Effects of algae on various parameters in piglets ^a^.

Algae	Treatment	Dose (CFU)	Pathogenic Challenge	Endpoints								Refs.
				GP	ND	FT/D	FS	GA	IS	FA	GM	
LAM	In feed; **T1**: maternal; **T2**: pw; **T3**: maternal and pw	sow diet: 1g/day LAM; weaner diet: 300 ppm LAM	*S.* Typhimurium PT12	- (T1: bw); + (T1: pw)	nm	+	+ (T1)	nm	+ (T1)	+	+ (T1, T3)	[197]
LAM-rich ^b^ SWE	In feed; sanitary conditions; **T1**: 18% CP + LAM, **T2**: 21% CP + LAM	300 ppm LAM	nm	NS	nm	+	nm	− (T2)	NS	nm	NS	[199]
LAM-rich ^c^ SWE	In feed, unsanitary conditions; **T1**: 18% CP diet + LAM, **T2**: 21% CP diet + LAM	300 ppm LAM	nm	+	nm	+	nm	nm	NS	nm	+	[199]
LAM-rich ^d^ SWE	In starter diet	300 ppm LAM	nm	+	nm	NS.	nm	nm	nm	+	+	[200]
Algae-derivedβ-glucans from dried *Euglena gracilis*	In starter diet	**L**: 54 mg/kg; **H**: 108 mg/kg	*E. coli* F18	NS	nm	+ (H: d3, 5)	nm	+	+	nm	nm	[201]
LAM, FUC	In starter diet; **T1**: LAM; **T2**: FUC	**T1**: 300 ppm LAM; **T2**: 240 ppm FUC	S. Typhimurium	NS	nm	NS.	− (T1: d14, 20); − (T2: d2,14)	nm	nm	+	+ (T2)	[202]
FUC-rich SWE	In feed	**L**: 125 ppm FUC; **H**: 250 ppm FUC	nm	NS	nm	+ (H)	nm	NS	nm	+ (H)	- (H)	[203]
LAM + FUC	In starter diet; **T1**: LAM + FUC **T2**: T1+ZnO	300 mg/kg LAM + 240 mg/kg FUC	nm	NS	+ (T1)	+ (T2)	nm	nm	nm	NS	NS	[204]
SWE including LAM (1.0 g), FUC (0.8 g), ash (8.2 g)	In lactation diet	10 g/day SWE	*E. coli* K88	+	nm	+	+	+	+	nm	nm	[205]
SWE including LAM (1.0 g), FUC (0.8 g), ash (8.2 g)	In starter and transition diet; **T1**: SWE; **T2**: T1 + ZnO	SWE: 300 ppm LAM + 240 ppm FUC; ZnO: 3.1 g/kg starter and 2.5 g/kg transition diet	nm	NS	+ (T1)	NS.	nm	nm	nm	nm	nm	[206]
LAM + FUC	In lactation diet; **T1**: LAM; **T2**: FUC; **T3**: LAM + FUC	**T1**: 1.0 g/day LAM; **T2**: 0.8 g/day FUC; **T3**: 1.0 g LAM and 0.8 g FUC/d	nm	+ (T2, T3, 26dpw)	nm	− (T2, T3)	nm	+ (T1, T3); − (T2)	+ (T1); − (T2)	nm	+ (T1, T3)	[196]
LAM, FUC	In starter diet; **T1**: LAM; **T2**: FUC; **T3**: LAM + FUC	**T1**: 300 ppm LAM; **T2**: 240 ppm FUC; **T3**: 300 ppm LAM + 240 ppm FUC	nm	+ (T1, T3)	nm	+ (T1, T3: 0-4dpw); + (T3: 0-21dpw)	nm	nm	nm	NS.	+ (T1, T3)	[207]
LAM, FUC	In starter diet; **T1**: LAM; **T2**: FUC; **T3**: LAM + FUC	**T1**: 300 ppm LAM; **T2**: 240 ppm FUC; **T3**: 300 ppm LAM + 240 ppm FUC	nm	nm	nm	+ (T1, T2)	nm	+ (T1, T2)	+ (T1)	+ (T1); +/− (T2)	+ (T1, T2)	[208]
LAM, FUC	In starter diet; **T1**, **T2**: LAM; **T3**: FUC; **T4**, **T5**: LAM + FUC	**T1**: 150 ppm; **T2**: 300 ppm; **T3**: 240 ppm; **T4**: 150 ppm LAM + 240 ppm FUC; **T5**: 300 ppm LAM + 240 ppm FUC	nm	+ (T2: ADG); + (T1-T3: G:F)	+	+ (T1, T2: 0-14dpw); + (T3: 14–21, 21–35 dpw)	nm	nm	nm	NS.	+ (T3)	[209]
SWE ^e^ from *Enteromorpha prolifera*	In starter diet	400 ppm	nm	+	nm	+	nm	+	+	nm	+	[210]
Macroalgae pro-duct (OFS)	In weaner diet (1–52 dpw)	15,000 ppm	nm	NS	nm	NS	nm	-	nm	nm	nm	[142]
Dried brown seaweed (*Ascophyllum Nodosum*)	In starter diet	**L**: 2.5 g/kg; **I**: 5.0 g/kg; **H**: 10.0 g/kg	nm	NS	nm	− (H)	nm	nm	nm	nm	NS	[211]

^a^ Effects +: positive effect; −: negative effect; NS: non-significant effect; nm: not measured. Only treatments including seaweed-derived polysaccharide extracts (SWEs) are stated, control treatments or other additives tested are not considered here. ppm: parts per million; L: low dose; I: intermediate dose; H: high dose; T: treatment; OFS: Macroalga-derived product containing multiple unspecified species of brown, green and red macroalgae (8.7% LAM, 3.7% FUC, 14% algin or alginate, 9.7% mannitol, 0.3% fucoxanthin and 13.7% rhamnose sulphate). ^b^ 44% FUC, 2.59% LAM, 13.5% alginates, 4.38% mannitol, 3.48% phlorotannins and 31.95% ash. ^c^ 690 g LAM/kg DM, 225 g FUC/kg DM, 16 g mannitol/kg DM, 42 g alginates/kg DM and 55 g ash/kg DM. ^d^ 653.2 g LAM/kg DM, 190 g FUC/kg DM, 5 g phlorotannin/kg DM, 51 g mannitol/kg DM, 40 g alginates/kg DM and 3.18 g ash/ kg DM. ^e^ 48% polysaccharides (40.6% rhamnose, 38.2% glucose, 9.3% glucuronic acid, 6.3% xylose and 5.6% galactose), 4.82% protein, 2.9% moisture, 17487.6 mg/kg calcium, 15362.7 mg/kg magnesium, 24262.4 mg/kg potassium, 3.93 mg/kg arsenium, 1.22 mg/kg plumbum, and 0.86 mg/kg cadmium.

**Table 8 animals-12-02585-t008:** Effects ^a^ of antimicrobial peptides (AMPs) on various parameters in piglets.

AMP	Dose	Replicates	Pathogenic Challenge	Endpoints ^b^						Refs.
				GP	FT/D	GA	EB	IS	GM	
WK3	2 mg/kg body weight (injection)	8 piglets	ETEC F4	+	+	+	nm	+	(+)	[224]
WK3	50 mg/kg feed	8 piglets	ETEC F4	+	+	+	nm	+	+	[226]
Cathelicidin-BF	0.6 mg/kg body weight (intraperitoneally)	3 pens with 6 piglets	nm	+	+	+	+	+	nm	[227]
KR-32		6 piglets	nm	nm	+	nm	nm	+	nm	[228]
Microcin J25	0,5; 1; 2 mg/kg feed	6 pens with 6 piglets	nm	+	+	nm	+	+	+	[223]
Porcine beta-defensin-2	1 mg/day orally	10 or 5 piglets	ETEC F4	+	nm	+	nm	+	+	[229]
Porcine beta-defensin-2	1; 5; 15 g /kg feed	3 pens with 10 piglets	nm	+ (day 1 to 28 post-weaning)	+ (day 1 to 28 post-weaning)	+ (day 28 post-weaning)	nm	nm	nm	[230]
Plectasin	60 mg/kg feed	6 piglets	nm	+ (day 1 to 21 post-weaning)	+	+	+	nm	+	[231]
Cecropin AD	400 mg/kg feed	8 piglets	ETEC F4	+	+	+	nm	+/−	+	[221]
Colicin E1	11; 16.5 mg/kg feed	8 piglets	ETEC F18	+ (high dose group)	+ (high dose group)	nm	nm	+ (high-dose group)	nm	[232]
Lactoferrin	1 g/kg feed	3 pens with 10 piglets		+ (day 1 to 28 post-weaning)	+ (day 1 to 28 post-weaning)	+ (day 28 post-weaning)	nm	nm	nm	[233]
Lactoferrin+ cecropin+ defensing+ and plectasin	2; 3 g/kg feed	84 pens with 11 piglets (5 farms)	nm	+ (day 1 to 32 post-weaning)	+ (day 1 to 32 post-weaning)	nm	nm	nm	nm	[234]
Lactoferrampin-lactoferricin	100mg/kg feed	10 piglets	nm	+	+	nm	+	+	nm	[235]
*Musca domestica* larvae AMP+ porcine defensin	400mg/kg feed	4 pens with 3 piglets	nm	+	+	nm	nm	NS	+ (day 28 post-weaning)	[236]

^a^ Effects. +: positive effect; −: negative effect; NS: Non-significant effect; nm: not measured. ^b^ GP: growth performance; ND: digestibility; FT/D: faecal trait/diarrhoea; GA: gut architecture/histology; EB: epithelial barrier; IS: immune system; GM: gut microbiology.

## Data Availability

Not Applicable.

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
