# Peer review of "Review on Preventive Measures to Reduce Post-Weaning Diarrhoea in Piglets"

_animals, 2022, doi:10.3390/ani12192585_

Round 1
Reviewer 1 Report
Weaning diarrhea of piglets is the biggest problem in the process of pig breeding. Especially in recent years, the ban on growth-promoting antibiotics and the reduction in the amount of medical zinc oxide have led to frequent weaning diarrhea in piglets. This article systematically starts from the aspects of feed ingredients, feed additives and feeding strategies, and summarizes the measures to reduce weaning diarrhea in piglets except adding antibiotics and medical zinc oxide. Overall, the manuscript is well written and the logical framework is well organized. This manuscript is proposed for publication in the journal Animals with minor revisions. The specific comments/suggestions are as follows:Line 67:Please remove extra punctuation.
1. Line 102:"Lopez-Galvez, et al. and Heo et al." font format inconsistent.
2. Line 113:Please indicate the full name of the first occurrence of the abbreviation.
3. Line 334:Please amend "cfu" in the table to "CFU".
4. Line 364: Please cite the expression uniformly.
5. Line 416: Note the font size of "E. coli".
6. Line 418: Use italics for bacterial expression.
7. Line 506-507: Note the font size before and after.
8. Line 547: Please remove the symbol after 'antibiotic'
9. Line 666: Use italics for bacterial expression.
10. Line 668: Use italics for bacterial expression.
11. Line 971:Please remove the symbol after 'lactobacilli'.
12. Line 972: Note the spacing between words.
13. Line 1002: Whether 'FUC' needs to be highlighted.
14. Line 1364: Use italics for bacterial expression.
15. Line 1436:Please indicate the source when using abbreviations.
16. Line 1453:Please indicate the full name of the first occurrence of the abbreviation.
17. Line 1462:Please modify the reference notation format.
18. Line 1069-1616:Please provide relevant references to support.
19. Line 2061:Please modify the reference notation format.
20. Line 2107:Please modify the reference notation format.
21. Line2115-2116:Please remove the symbols after pH and temperature.
Author Response
REVIEWER 1
Weaning diarrhea of piglets is the biggest problem in the process of pig breeding. Especially in recent years, the ban on growth-promoting antibiotics and the reduction in the amount of medical zinc oxide have led to frequent weaning diarrhea in piglets. This article systematically starts from the aspects of feed ingredients, feed additives and feeding strategies, and summarizes the measures to reduce weaning diarrhea in piglets except adding antibiotics and medical zinc oxide. Overall, the manuscript is well written and the logical framework is well organized. This manuscript is proposed for publication in the journal Animals with minor revisions.
The specific comments/suggestions are as follows:
Line 67:Please remove extra punctuation.
I am not sure what is meant by this.
- Line 102:"Lopez-Galvez, et al. and Heo et al." font format inconsistent.
Done
- Line 113:Please indicate the full name of the first occurrence of the abbreviation.
Line 113 says: pigs or weaner or weaners or porcine or swine) AND (weaning or postweaning or post-
So I am not sure what this comment refers to.
- Line 334:Please amend "cfu" in the table to "CFU".
Done
- Line 364: Please cite the expression uniformly.
Done
- Line 416: Note the font size of "E. coli".
Done
- Line 418: Use italics for bacterial expression.
Done
- Line 506-507: Note the font size before and after.
Done
- Line 547: Please remove the symbol after 'antibiotic'
In my opinion the symbol should stay because it refers to antibiotic-treated
- Line 666: Use italics for bacterial expression.
Done
- Line 668: Use italics for bacterial expression.
Done
- Line 971:Please remove the symbol after 'lactobacilli'.
Done
- Line 972: Note the spacing between words.
That is done automatically by Word.
- Line 1002: Whether 'FUC' needs to be highlighted.
It has been defined on line 935 (R1 version).
- Line 1364: Use italics for bacterial expression.
I have changed to lowercase letters and no italics.
- Line 1436:Please indicate the source when using abbreviations.
Done
- Line 1453:Please indicate the full name of the first occurrence of the abbreviation.
Done
- Line 1462: Please modify the reference notation format.
Done
- Line 1069-1616:Please provide relevant references to support.
I assume the reviewer meant line 1609 to 1616. Done.
- Line 2061:Please modify the reference notation format.
Done
- Line 2107:Please modify the reference notation format.
Done
- Line2115-2116:Please remove the symbols after pH and temperature.
In my opinion, they are correct because all refer to stable.
nderstand the comment.

Reviewer 2 Report
This manuscript represents a complete state of the art review regarding the most recent preventive measures assessed to prevent post-weaning diarrhoea in piglets. Although other reference reviews can be found in the literature, probably no one represent so exhaustive review on the topic integrating most of the alternatives to antimicrobials in pig production. Authors make a noteworthy work reviewing the most recent publications, articulating the writing in rational sections (with nice summarizing tables), and giving a critical view of the strengths and drawbacks of each kind of compounds or strategies. Undoubtedly this is a work that deserve publication, probably turning in a reference paper on this topic in the short term.
The manuscript may be results too long for a standard publication in a Journal, however it is justified considering that is the systematic approach to the topic what actually represents the main strength of the paper. If the length is not a problem for the journal (Editor) I would recommend to maintains its extension.
One important formal aspect in the manuscript is the inaccurate use of microbial taxonomy and particularly italics for some bacterial names. Below you will find some comments, however you will need to review exhaustively the whole paper as there are multitude of inaccuracies.
Additionally he manuscript needs to be improved in some minor details (see comments below).
L69-71. This sentence is not clear and probably no needed here. Remove?
Introduction: A more clear explanation of the link between medical levels of ZnO and selection of AMR for the readers would be advisable.
L98-99. Review writing of this sentence. It is not clear.
195-198. From my point of view this strategy would better fall within the previous sub-section of modes of action (Establishment of a robust gut microbiota).
L247. May be also to consider the interaction with the quorum sensing mechanisms of pathogens.
296. Italics for Bacillus
L316 Italics for Lactobacillus
L320-321-Italics. Review italics for microbial names along the manuscript.
Table 2. It would be recommendable to specify the route of administration (p.e. in the feed (dry, liquid, fermented..), in the water, oral dose…).
339. Bifidobacterium should read in italics.
340 Bifidobacteria should read not in italics.
352 Typhimurim should read not in in italics (it is a serovar name). Review along the whole manuscript.
361. Salmonella in italics.
L502. Review font size for some words along the manuscript. (p.e. diarrhoea here).
L572. bifidobacteria and lactobacilli
L615-617. Would not this kind of products fall within the “post biotic concept”?
L700-703. It would be also interesting to consider possible deleterious effects of combining pro and prebiotics.
L971-Prevotella (Capitals and italics)
1006. Review font size.
L1139 Review break line
L1133-1141. This paragraph results difficult to read. It is not clear if soluble fiber is or not beneficial for the animal. Please review/complete writing.
L1510-1525. I do not see this paragraph regarding organic acids under this title (Dietary SCFA and MCFA). It can create confusion to the readers as organic acids are different from fatty acids. I would suggest removing this paragraph or reconsider writing.
L1621. Delete duplicity.
L1363. Delete comma.
L1667. Considering the length of the manuscript (too long for a journal) I would suggest removing this section (Creep-feeding) as it is a little bit out of the scope of the review. It could be considered as a management strategy (formerly excluded from the review).
L2004. lactobacilli do not need capital letter.
L1899. Bacteriophages section seems too long (too much descriptive). It could be summarized a little bit.
L2144 Postbiotics?
Table 9 and Table 10 are shown after the references section, but I guess they are not supplementary material. Are they?
Author Response
REVIEWER 2
This manuscript represents a complete state of the art review regarding the most recent preventive measures assessed to prevent post-weaning diarrhoea in piglets. Although other reference reviews can be found in the literature, probably no one represent so exhaustive review on the topic integrating most of the alternatives to antimicrobials in pig production. Authors make a noteworthy work reviewing the most recent publications, articulating the writing in rational sections (with nice summarizing tables), and giving a critical view of the strengths and drawbacks of each kind of compounds or strategies. Undoubtedly this is a work that deserve publication, probably turning in a reference paper on this topic in the short term.
The manuscript may be results too long for a standard publication in a Journal, however it is justified considering that is the systematic approach to the topic what actually represents the main strength of the paper. If the length is not a problem for the journal (Editor) I would recommend to maintains its extension.
One important formal aspect in the manuscript is the inaccurate use of microbial taxonomy and particularly italics for some bacterial names. Below you will find some comments, however you will need to review exhaustively the whole paper as there are multitude of inaccuracies.
Checked.
Additionally he manuscript needs to be improved in some minor details (see comments below).
L69-71. This sentence is not clear and probably no needed here. Remove?
This sentence is very important because indicates what studies are in focus/included here: the response parameter of diarrhoea/faecal score and the time of sampling i.e., approx. 2 weeks post-weaning. This is in contrast to studies where only surrogates for impaired health/diarrhoea are included or where samples are taken at around 4 weeks post weaning (which is not uncommon).
Introduction: A more clear explanation of the link between medical levels of ZnO and selection of AMR for the readers would be advisable.
Done
L98-99. Review writing of this sentence. It is not clear.
Done
195-198. From my point of view this strategy would better fall within the previous sub-section of modes of action (Establishment of a robust gut microbiota).
I have included this strategy in this section because the studies referred to investigate several parameters to gut development/maturity besides microbiota composition. So, I would like to keep it here.
L247. May be also to consider the interaction with the quorum sensing mechanisms of pathogens.
Added
- Italics for Bacillus
Done
L316 Italics for Lactobacillus
Done
L320-321-Italics. Review italics for microbial names along the manuscript.
Done
Table 2. It would be recommendable to specify the route of administration (p.e. in the feed (dry, liquid, fermented..), in the water, oral dose…).
We consider that the table is already rather big with a lot of information. We state the dosis provided, which is the most important information, and the route, when not clear, can be seen in the original paper. So, we have not added the route.
- Bifidobacteriumshould read in italics.
Done
340 Bifidobacteria should read not in italics.
Done
352 Typhimurim should read not in in italics (it is a serovar name). Review along the whole manuscript.
Done
- Salmonellain italics.
Done
L502. Review font size for some words along the manuscript. (p.e. diarrhoea here).
Done
L572. bifidobacteria and lactobacilli
Done
L615-617. Would not this kind of products fall within the “post biotic concept”?
I agree. I have added a sentence on line 661-663.
L700-703. It would be also interesting to consider possible deleterious effects of combining pro and prebiotics.
That is true but I have not found data on that aspect in the literature reviewed here.
L971-Prevotella (Capitals and italics)
Done
- Review font size.
Done
L1139 Review break line
Done
L1133-1141. This paragraph results difficult to read. It is not clear if soluble fiber is or not beneficial for the animal. Please review/complete writing.
Modified
L1510-1525. I do not see this paragraph regarding organic acids under this title (Dietary SCFA and MCFA). It can create confusion to the readers as organic acids are different from fatty acids. I would suggest removing this paragraph or reconsider writing.
I agree. Modified
L1621. Delete duplicity.
Done
L1363. Delete comma.
Done
L1667. Considering the length of the manuscript (too long for a journal) I would suggest removing this section (Creep-feeding) as it is a little bit out of the scope of the review. It could be considered as a management strategy (formerly excluded from the review).
The intake of creep feed is consider crucial for the risk of post-weaning diarrhoea, and although somewhat management, it is it is related to feed. We think it would be a pity to delete this section, also because we present valuable information, like that the optimal level of creep feed intake is not clear, etc.
L2004. lactobacilli do not need capital letter.
Done
L1899. Bacteriophages section seems too long (too much descriptive). It could be summarized a little bit.
Done
L2144 Postbiotics?
It is correct as it is.
Table 9 and Table 10 are shown after the references section, but I guess they are not supplementary material. Are they?
No, they are not. The tables have been placed by the journal but for the final version they will be moved to their correct position.

Reviewer 3 Report
Dear Authors,
I expect it was a long and hard work to prepare this manuscript, the amount of work is noticeable. I am impressed, however I don't know what was your real goal - to write a manuscript or the guide: what to do/how to treat or prevent PWD. This manuscript is too long and it is very difficult to read/study it.
Due to various aspects mentioned there I don't recommend shortening it but you have to divide it into 2-3 parts. You have to decide according what rules to do it, perhaps basis on your Table 1? Because of multifactorial character of this disease we can talk about: direct and indirect /specific and unspecific prevention; we can present factors influencing gut microbiom (direct and indirect); we can present factors influencing development and maturation of intestine/gastrointestinal tract; we can present factors inducing specific immunity or just immunity. Presenting by you solutions sometimes overlap what makes difficult this description, but don't worry about it and sometimes you can repeat something. Depending on the chosen strategy you can cite and complete your previous part of the study.
And in the end a few comments and addition;
- you have to rewrite /rebuilt? this manuscript or just ask the editorial office to make it a special issue;
- check legends below your tables, below the Tab.2 lack of description what "nm" means, instead of it we have "na" which we can find in Tab.6, in Tab. 5 suddenly we have "NM" probably instead of nm;
- for me sIgA means secretory IgA not serum IgA, however you explained it in the text, but I think it is wrong description;
- you wrote about using immunoglobilines obtained from pig plasma and cows' colostrum, if you decided to introduce some changes in the manuscript I think it is worth to mention about using yolk immunoglobulines (IgY);
- when you wrote about oral immunisation of sows to obtain specific Ig in colostrum - in this aspect rather we think about rearing stronger and healthier piglets and thanks to it we can expect lower danger of PWD appearance in weaned piglets.
Author Response
REVIEWER 3
I expect it was a long and hard work to prepare this manuscript, the amount of work is noticeable. I am impressed, however I don't know what was your real goal - to write a manuscript or the guide: what to do/how to treat or prevent PWD. This manuscript is too long and it is very difficult to read/study it.
Due to various aspects mentioned there I don't recommend shortening it but you have to divide it into 2-3 parts. You have to decide according what rules to do it, perhaps basis on your Table 1? Because of multifactorial character of this disease we can talk about: direct and indirect /specific and unspecific prevention; we can present factors influencing gut microbiom (direct and indirect); we can present factors influencing development and maturation of intestine/gastrointestinal tract; we can present factors inducing specific immunity or just immunity. Presenting by you solutions sometimes overlap what makes difficult this description, but don't worry about it and sometimes you can repeat something.
Depending on the chosen strategy you can cite and complete your previous part of the study.
And in the end a few comments and addition;
- you have to rewrite /rebuilt? this manuscript or just ask the editorial office to make it a special issue;
XXX
I find it difficult to reorganize it in parts, for example according to Table 1 because the same alternative can have several modes of action.
I have organized a little bit the order so that first additives, then dietary ingredients, followed by creep feed and at the end a non-feeding strategy, i.e., vaccines
- check legends below your tables, below the Tab.2 lack of description what "nm" means, instead of it we have "na" which we can find in Tab.6, in Tab. 5 suddenly we have "NM" probably instead of nm;
Done
- for me sIgA means secretory IgA not serum IgA, however you explained it in the text, but I think it is You are right, it was not correct. It has been modified.
- you wrote about using immunoglobilines obtained from pig plasma and cows' colostrum, if you decided to introduce some changes in the manuscript I think it is worth to mention about using yolk immunoglobulines (IgY);
You have a point but we have concentrated on milk proteins, not on proteins in general. IN that case, other products like blood plasma and other products should also be included. So, we do not want to make the paper longer.
- when you wrote about oral immunisation of sows to obtain specific Ig in colostrum - in this aspect rather we think about rearing stronger and healthier piglets and thanks to it we can expect lower danger of PWD appearance in weaned piglets.
A better immunity in the piglets, via higher Ig, should also contribute to more robust and healthier animals, so there is contradiction in that.
I am not sure, I understand the comment.
